# Evolutionary pathways of repeat protein topology in bacterial outer membrane proteins

Meghan Whitney Franklin[1], Sergey Nepomnyachyi[2,3], Ryan Feehan[1], Nir Ben-Tal[2], Rachel Kolodny[3], Joanna SG Slusky[1,4]*

[1]Center for Computational Biology, University of Kansas, Kansas, United States; [2]Department of Biochemistry and Molecular Biology, George S. Wise Faculty of Life Sciences, Tel Aviv University, Ramat Aviv, Israel; [3]Department of Computer Science, University of Haifa, Haifa, Israel; [4]Department of Molecular Biosciences, University of Kansas, Kansas, United States

**Abstract** Outer membrane proteins (OMPs) are the proteins in the surface of Gram-negative bacteria. These proteins have diverse functions but a single topology: the β-barrel. Sequence analysis has suggested that this common fold is a β-hairpin repeat protein, and that amplification of the β-hairpin has resulted in 8–26-stranded barrels. Using an integrated approach that combines sequence and structural analyses, we find events in which non-amplification diversification also increases barrel strand number. Our network-based analysis reveals strand-number-based evolutionary pathways, including one that progresses from a primordial 8-stranded barrel to 16-strands and further, to 18-strands. Among these pathways are mechanisms of strand number accretion without domain duplication, like a loop-to-hairpin transition. These mechanisms illustrate perpetuation of repeat protein topology without genetic duplication, likely induced by the hydrophobic membrane. Finally, we find that the evolutionary trace is particularly prominent in the C-terminal half of OMPs, implicating this region in the nucleation of OMP folding.
DOI: https://doi.org/10.7554/eLife.40308.001

*For correspondence: slusky@ku.edu

## Introduction

Outer membrane beta barrels (OMBBs) have a remarkably homogeneous architecture. All known OMBBs in Gram-negative bacteria, save one (*Dong et al., 2006*), are right-handed, up-down beta barrels with the N and C termini of the barrel remaining on the membrane face from which they are inserted. Yet, these proteins carry out all the functions necessary for the interface between the cell and its environment: adhesion, various specific and nonspecific forms of import and efflux, pilus formation, and proteolysis.

Evolutionary diversification yields two main categories of changes that modulate function among OMBBs: changes in the barrel's girth and changes in the location and identity of functional amino acids. The barrel's girth is a function of strand number. Bacterial OMBBs each have an even number of strands between 8 and 26. The addition of each hairpin (two strands connected by a loop) imparts an extra 1.66 Å to the radius (*Franklin et al., 2018a*) in the same way that the addition of an extra two planks to a cask widens the barrel's radius by a consistent amount. Changing the radius of a barrel alters pore specificity to accommodate or reject molecules for passage based on size. Widening or narrowing the barrel also changes the location of the loops, which makes for better adhesion or pilus formation. OMBB function can also be diversified in the location or identity of particular amino acids. Diversity of amino acids allows similarly sized pores to have different chemical specificity. Moreover, chemically potent residues impart proteolytic or adhesive capabilities.

Although there are few instances in which the membrane barrel topology was converged upon independently (*Franklin et al., 2018a*), most OMBBs are homologous as shown by sequence alignments (*Remmert et al., 2010*; *Reddy and Saier, 2016*). Like all diversification events, the diversification that maintained this common fold while accommodating diverse function occured through a combination of amplification, recombination, and the accretion of mutations. The placement and identity of functional amino acids is most commonly attributed to the accretion of mutations. Conversely, previous work has clarified that the original repeat unit of the beta hairpin or double hairpin has been amplified to create barrels of different strand numbers (*Remmert et al., 2010*). That the OMBBs are all composed of different numbers of tandem hairpins is the basis of early studies demonstrating that OMBBs are a form of repeat protein (*Neuwald et al., 1995*).

In general, repeat proteins are believed to be generated by duplication and recombination within a single gene and less commonly by polymerase or strand slippage of DNA hairpins (*Marcotte et al., 1999*). Such gene and motif duplication is a common mechanism for generating biological diversity and has occurred throughout evolution, resulting in larger proteins and increasing the diversity of protein function (*Ohno, 1970*; *Zhang, 2003*). Gene duplication is known to be especially prevalent in membrane proteins (*Shimizu et al., 2004*). It is noteworthy that the lack of exons in prokaryotes eliminates the possibility of complex editing and recombination events as a mechanism for increasing biological diversity.

The diversification of the OMBBs depends in part on their relative rates of duplication and mutation. Though the relative rates of these two pathways vary by organism, overall duplications are more common than substitution mutations (*Katju and Bergthorsson, 2013*). The mutation rate in *E. coli* is $5.4 \times 10^{-10}$ substitutions per base pair per replication (*Drake et al., 1998*), while the duplication rate in the closely related *Salmonella* is $2 \times 10^{-3}$ to $4.6 \times 10^{-6}$ duplications per gene per generation (*Reams et al., 2010*). At approximately $1 \times 10^3$ base pairs per 300-residue OMBB protein, that rate is $10^{-7}$ mutations per gene (including silent mutations). Therefore, the rate of a protein acquiring a single mutation is less than the rate at which a protein will acquire a duplication. However, which of the mutations or duplications are fixed is determined by the benefit, detriment, or neutrality of the diversification event.

Here we explore OMBB diversification through structural similarities and differences among evolutionarily related OMBBs. To do this, we need (1) structural information and (2) a sensitive alignment method. With respect to structural information, we need an OMBB's strand number and strand/loop boundaries. With respect to the alignment method we need a sensitive method which identifies evolutionarily related OMBBs and which accurately matches related parts. For structural studies, the accuracy of structure prediction is limited. A recent study reports 69% accuracy in correct topology prediction (*Hayat et al., 2016*). While this accuracy is impressive, it is insufficient for our purposes. Therefore to address our structural needs we used OMBBs with experimentally-solved structures, so that we have the true strand number and strand identity. This constitutes a major difference between this work and previous studies which relied on structure prediction (*Remmert et al., 2010*; *Reddy and Saier, 2016*). We amplified the signal for each structurally solved protein by creating a Hidden Markov Model (HMM) of each protein and its homologues. HMMs are statistical models that represent the sequences of the protein and many of its homologs. It is these HMMs which are then aligned. Using structurally solved proteins to generate HMMs for alignment gives us detailed structural knowledge along with the statistical power of thousands of sequences.

By correlating this structural data with evolutionary relationships, we find a strand-number-based pathway of evolution from 8- to 16-stranded barrels and then to 18-stranded barrels. Moreover, we find evidence that strand-number diversification can occur through mutation, challenging the longstanding view that repeat topology arises from repeating DNA alone. Finally, we find that the C-terminal half of OMBBs shows a particularly strong evolutionary trace signal, which may be relevant to understanding the OMBB folding pathway.

## Results

### Several groups of OMBBs

We compiled a data set of 138 OMBBs at <85% sequence similarity, 118 of which share <50% sequence similarity (*Franklin et al., 2018a*; *Franklin and Slusky, 2018b*). To determine evolutionary

relationships, we selected homologues to the structurally solved proteins from a database of 279,715 nonredundant sequences and generated hidden Markov model (HMM) profile alignments. The HMMs determine the likelihood of a relationship using a probabilistic model of the multiple sequence alignment and are therefore representative of all homologous sequences in the profile. The results were given as sequence alignments and a score called an E-value, which is the expected probability that there is false positive with a similarity greater than the given score (*Söding, 2005*). These results were mapped into a network model with proteins as nodes and alignments as connections drawn based on the magnitude of the E-value.

In this paper, we focus exclusively on the largest group found at an E-value of $10^{-3}$, which we call the prototypical beta-barrels. We use the E-value $\leq 10^{-3}$ because that value is the most widely used to infer homology, that is evolution from a common ancestor, (*Pearson, 2013*) and because it cleanly distinguished the prototypical barrels from other groups of OMBBs. The other groups of OMBBs, such as the efflux pumps, assembly barrels, and Fim/usher proteins, are described more fully elsewhere (*Franklin et al., 2018a*).

The majority (71% or 97/138) of OMBBs fall into this single, highly interconnected group and their HMMs collectively represent 50,832 sequences. An interactive version of the prototypical network, allowing visual inspection of the underlying sequence and structural similarity, is available online (http://cytostruct.info/rachel/protos/index.html). The interactive, online network has a scroll bar so that the user can change the E-value cutoff.

Our group of structurally solved prototypical OMBBs is composed of multiple subclusters (*Figure 1* and *Video 1*). The subclusters are organized by strand number; most barrels that share a strand number are more closely related than barrels that differ in the number of strands. The relationships between the subclusters articulate evolutionary relationships between the strand numbers.

## Connections between and among barrels of different strand numbers

Overall, we find that as barrels increase in strand number there is a slight increase in the average amino acid volume in the strands (*Figure 2—figure supplement 1*), likely because larger barrels would require larger amino acids to enforce the smaller curvature of the larger barrel's circumference. However, such differences do not mute the relationships between barrels of different sizes.

As discussed above, it is generally understood that a change in strand number would be caused by an amplification event—most likely duplication—and a change in sequence without a change in strand number would most likely be caused by the accretion of mutations. In order to assess the relative frequency of diversifying and then fixing amplification events versus mutation events, we compared the frequency of an alignment between barrels of different strand numbers to the frequency of alignments of barrels with the same strand number. We assessed both the quantity and quality of the alignments between barrels of different strand numbers and among barrels of the same strand number. In order to not over preference the frequency and quality of the alignments among barrels of the same size, we narrowed our dataset to the 53 barrels of only 25% sequence similarity. The equivalent of *Figure 1* at 25% sequence similarity is shown in *Figure 1—figure supplement 1*.

With respect to the quality of alignments, we find that E-values are always lower among barrels of the same strand number than between barrels of different strand numbers (*Figure 2—source data 1*). With respect to the quantity of alignments, there is, in all cases, a higher frequency of alignments between strands of the same size barrel than alignments between different sized barrels (*Figure 2*). The close relationships among barrels of the same size suggest that the different quantity of structures for each barrel size in the PDB may not have a strong biasing effect on our data as barrels of the same size tend to be related. Moreover, although duplication events are known to be more frequent than mutation events, the high frequency and high quality of the alignments between barrels of the same size strongly suggest that for OMBBs, mutation events are much more frequently fixed than duplication events.

## Internal repeats

In order to better understand the repeat nature of the evolution from one strand number to another for all barrel types described above, alignments of sequence similar strands were found internally by allowing each barrel to self-align. We will refer to internal alignments as internal repeats and to external alignments as alignments.

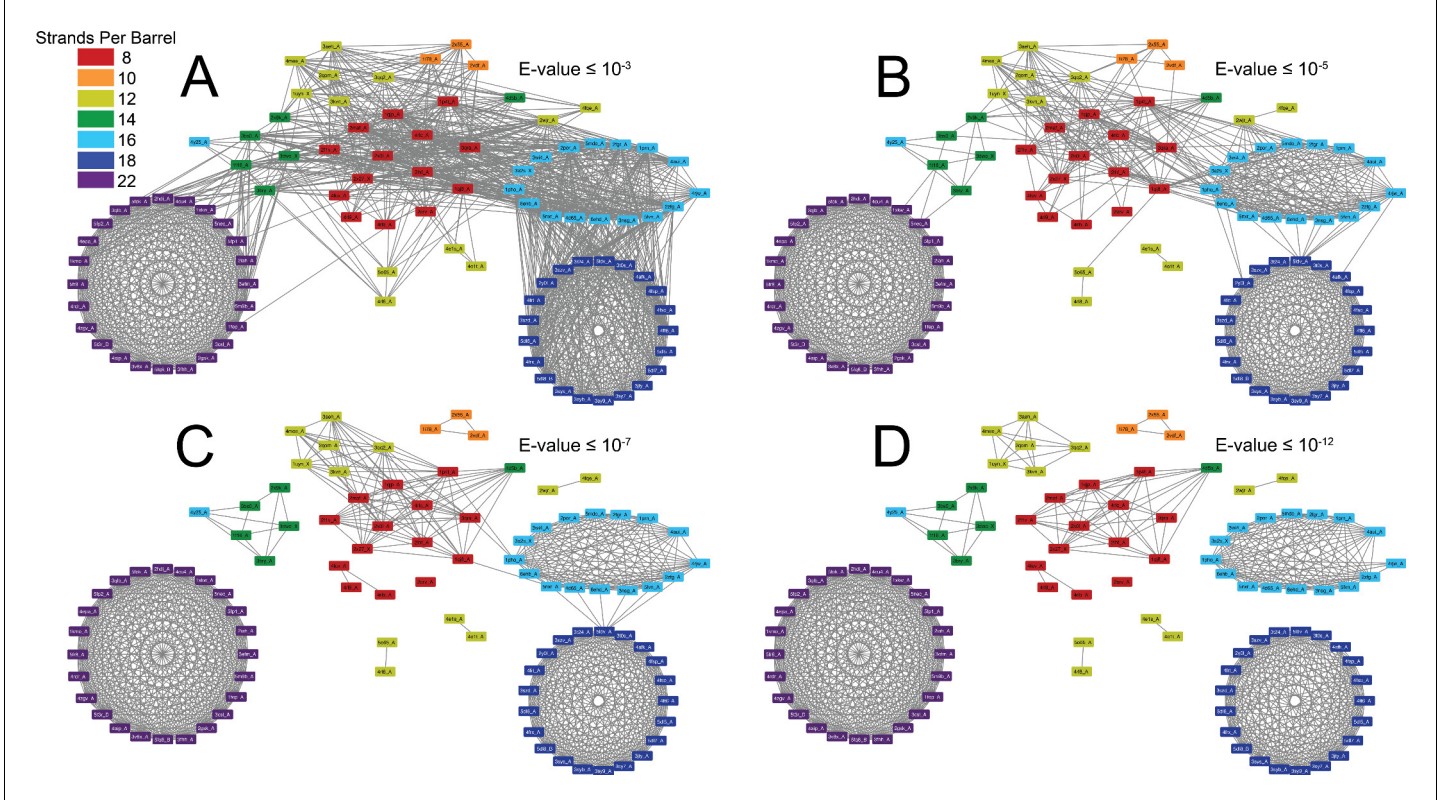

**Figure 1.** The prototypical barrels at E-values $\leq 10^{-3}$ to $10^{-12}$ colored by number of strands. Each edge represents sequence similarity between two nodes (barrels) with an E-value less than or equal to the subcaption. Multiple edges between the same pair of nodes have been removed for clarity; the edge with the lowest E-value was kept for visualization. (**A**) E-values $\leq 10^{-3}$, (**B**) E-values $\leq 10^{-5}$, (**C**) E-values $\leq 10^{-7}$, (**D**) E-values $\leq 10^{-12}$. A gif toggling through the E-values is shown in **Video 1**. The equivalent of **Figure 1** at 25% sequence similarity is shown in **Figure 1—figure supplement 1**. **Supplementary file 1** lists the list the PDBs of the full OMBB dataset. **Figure 1—source data 1** show all the alignments.

DOI: https://doi.org/10.7554/eLife.40308.002

The following source data and figure supplement are available for figure 1:

**Source data 1.** All alignment data.
DOI: https://doi.org/10.7554/eLife.40308.004

**Figure supplement 1.** Prototypical barrels at 25% sequence similarity at an E-value $\leq 10^{-3}$ to $10^{-12}$ colored by number of strands.
DOI: https://doi.org/10.7554/eLife.40308.003

Internal repeats show the duplications that lead to a full-length protein. All internal repeats within prototypical barrels are hairpin shifts or double hairpin shifts. A hairpin shift is where a set of strands align such that the first two strands are shifted over by two strands in the alignment; for example, a 7-stranded hairpin shift would be an alignment between strands one to seven and strands three to nine. A double hairpin shift is where a set of strands aligns such that the aligned strands of the two alignments are shifted over by four strands with respect to each other.

Internal repeats are identified in 39% of the prototypical barrels and are more characteristic of the smaller β-barrels than the larger ones (**Figure 3A** with a breakdown by type of internal repeat in **Figure 1—figure supplement 1A and B**). Internal repeats are generally quite robust –

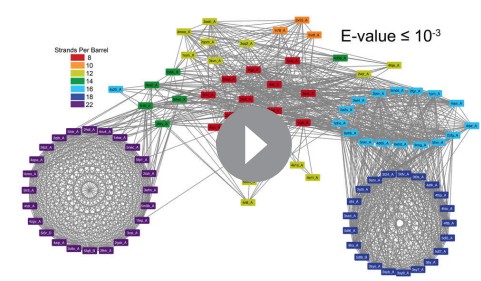

**Video 1.** Gif of the connection network showing the strength of the relationship between outer membrane barrels with different strand numbers. Gif toggles from E-value $\leq 10^{-3}$ to E-value $\leq 10^{-12}$.
DOI: https://doi.org/10.7554/eLife.40308.005

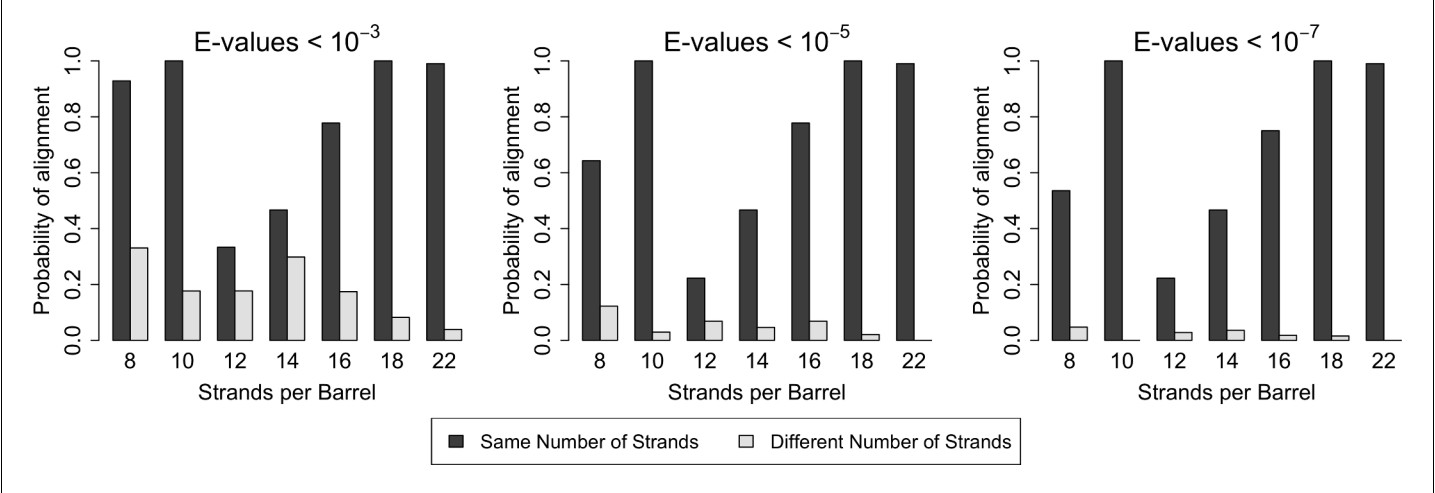

**Figure 2.** Similarity is higher among prototypical barrels with the same number of strands. Probability of barrels aligning with other prototypical barrels of the same number of strands compared to the probability of barrels aligning with prototypical barrels with different numbers of strands. Only prototypical OMBBs with <25% sequence similarity were considered. Alignments of OMBBs shown at E-values < $10^{-3}$ (left), E-values < $10^{-5}$ (center), E-values < $10^{-7}$ (right). If multiple alignments between two proteins existed, we used only the alignment with the smallest E-value. Probability was calculated as the number of alignments in each group divided by the possible combinations of barrel alignments for each group. Moreover, we find that E-values are always lower among barrels of the same strand number than between barrels of a different strand number (*Figure 2—source data 1*). Although barrel residues increase in size as they increase in strand number (*Figure 2—figure supplement 1*) such differences do not mute the relationships between barrels of different sizes.

DOI: https://doi.org/10.7554/eLife.40308.006

The following source data and figure supplement are available for figure 2:

**Source data 1.** Alignments between barrels of the same strand number have lower E-values.
DOI: https://doi.org/10.7554/eLife.40308.022

**Figure supplement 1.** Average side chain volume by the size of the barrel in strands.
DOI: https://doi.org/10.7554/eLife.40308.007

even at an E-value of $\leq 10^{-5}$, 36% of the prototypical barrels maintain an internal repeat. A full list of internal alignments identified for all 138 proteins can be found in the supplement (*Figure 2—source data 1*). Here, we enumerate the patterns that we find in the prototypical barrels (*Figure 3* and *Figure 3—figure supplement 1*).

The 8-stranded barrels only have double hairpin shifts (*Figure 3C*), while the 12-, 14-, and 16-stranded barrels have single hairpin shifts (*Figure 3—figure supplement 1C and D*, and *Figure 3D*). No internal repeats are detected in the 10- or 18-stranded barrels. Regardless of the internal repeats that do exist, the 8-, 12-, 14-, and 16-stranded barrels all contain at least one strand that does not contribute to an internal repeat. In the 8-stranded barrels, the fourth and eighth (last) strands are not found in any repeat. In the 12-, 14-, and 16-stranded barrels, the unaligned strands are the first few strands (strand one, strands one to three, and strands one to five, in the 12-, 14-, and 16-stranded barrels respectively).

## Alignments

While internal repeats within a protein may reveal the duplications that lead to a full length protein, alignments between proteins can tell us how the proteins are evolutionarily linked to each other. The alignments between barrels of different strand numbers in the prototypical group (*Figure 4*) suggest a potential evolutionary pathway. In this section, we describe these alignments in two steps: (1) how the eight-stranded barrels align with the 10-, 12-, 14-, and 16-stranded barrels; and (2) how 14-stranded barrels align with 22-stranded barrels and how 16-stranded barrels align with 18-stranded barrels. The same strands that align within the first step are included in the alignment in the second step. So, although 8-stranded barrels do not align with 18- or 22-stranded barrels, many of the strands are passed through. This means that strands from 8-stranded barrels that align with the 16-stranded barrels are included in the strands in the 16-stranded barrels that align with the 18-

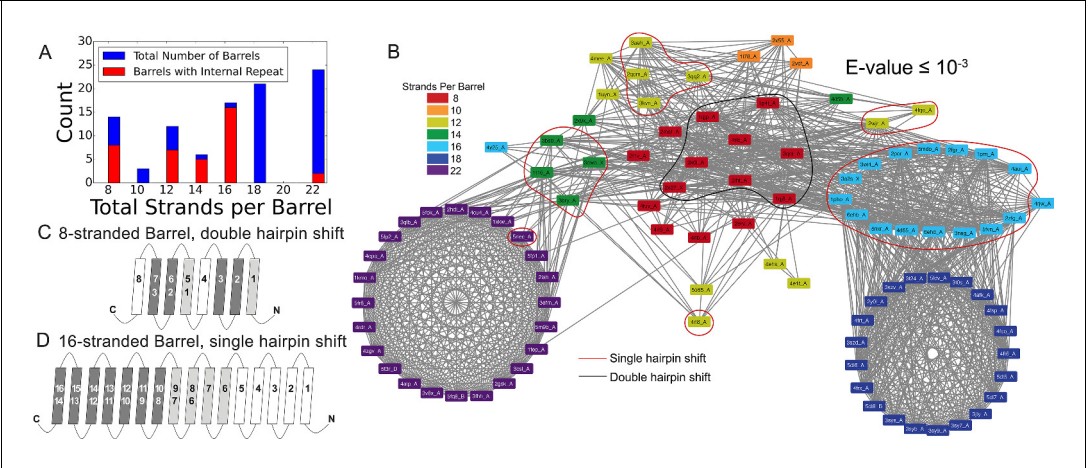

**Figure 3.** Internal repeats of the prototypical barrels. (**A**) Distribution of all prototypical barrels with an internal repeat. The blue bars represent the total counts, while the red bars represent the counts with an internal repeat. (**B**) Internal repeats are identified for the barrels that are circled. Red circles indicate a single hairpin shift; black indicates a double hairpin shift (See text for definition). (**C,D**) Strands are shown going right to left, N to C, following the orientation in the membrane, with up indicating extracellular and down indicating intracellular. Dark grey strands indicate that all barrels of that size share those strands in the repeat; light grey strands indicate that only some of the barrels have those strands involved in the internal repeat. White strands indicate that these strands are never observed to participate in an internal repeat. The top row of numbers represents the strand number and the bottom row of numbers represents the strand numbers that align with the top numbers in the internal repeat. (**C**) 8-stranded barrels, (**D**) 16-stranded barrels. *Figure 3—figure supplement 1* shows the breakdown of *Figure 3A* by type of repeat as well as the repeat patterns for 12- and 14-stranded barrels. *Figure 3—source data 1* show all internal alignments.

DOI: https://doi.org/10.7554/eLife.40308.008

The following source data and figure supplement are available for figure 3:

**Source data 1.** All internal repeat alignments.

DOI: https://doi.org/10.7554/eLife.40308.010

**Figure supplement 1.** Additional details regarding internal repeats.

DOI: https://doi.org/10.7554/eLife.40308.009

stranded barrels. Similarly, the strands in the 8-stranded barrels that align with 14-stranded barrels are included in the strands in the 14-stranded barrel that align with the 22-stranded barrels. Examples of sequence alignments between barrels of different sizes and the lengths of the alignments are shown in the supplement (*Figure 4—figure supplement 1*, *Figure 4—source data 1*).

## Step 1

The alignments between 8-stranded barrels to 10-, 12-, 14-, and 16-stranded barrels represents the first step of strand number diversification. The eight-stranded barrels are at the center of the network (*Figure 1A–C*) and have similar alignments to 10-, 12-, 14-, and 16-stranded barrels. When the 8-stranded barrels align with the other barrels, the alignment includes at least two, and often as many as all 8, of the strands (*Figure 4—figure supplement 1*, *Figure 4—source data 1*). These alignments are positioned at the C-terminus such that the last strands of both barrels align, the penultimate strands align, the third-to-last strands align, etc. We find a C-terminal alignment in 165/186 alignments involving an 8-stranded barrel. These alignments represent relationships among 22,176 sequences included in the HMMs. However, all 8 strands of the 8-stranded barrel are not always involved in these alignments – although 145/186 alignments involve the last strand of both barrels, just 41/186 cases of this type of alignment contain all eight strands (representing relationships among 19,460 and 13,736 sequences included in the HMMs, respectively).

Moreover, we find alignments between 10- and 12-stranded barrels (*Figure 4—figure supplement 1E*), 12- and 14-stranded barrels (*Figure 4—figure supplement 1F*), and 12- and 16-stranded barrels (*Figure 4—figure supplement 1G*). We are unable to differentiate, however, between scenarios in which the alignments between two such barrels resulted from the evolution from one barrel to another and those in which they both evolved from an original 8-stranded barrel. Essentially, all alignments between barrels in the second tier of *Figure 4* follow the pattern of C-terminal alignment

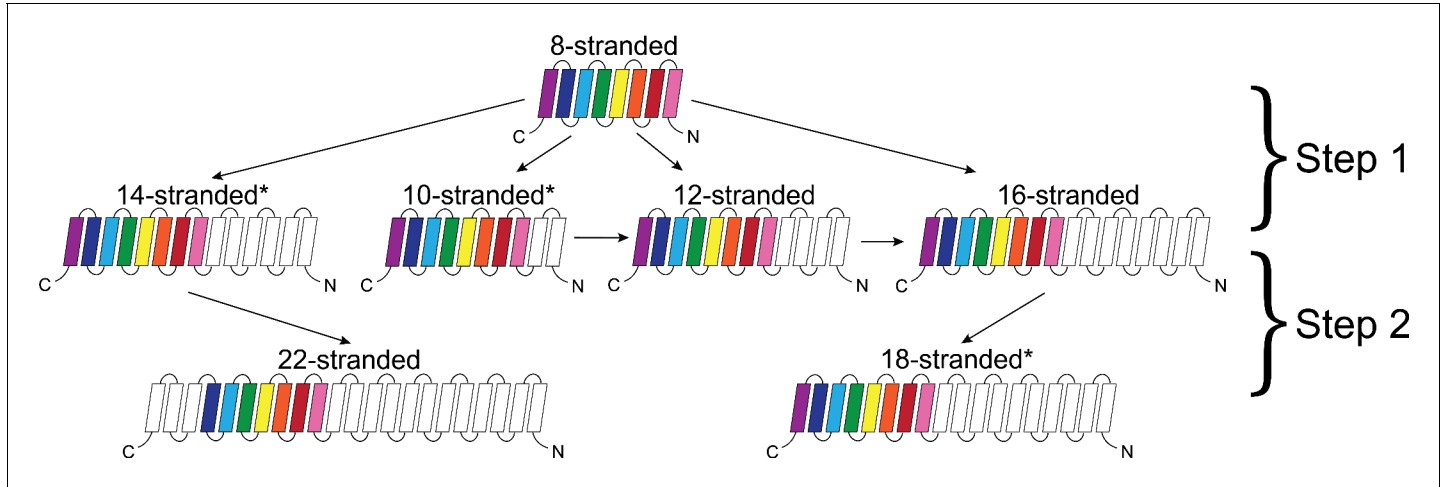

**Figure 4.** Dominant mode of persistence of the eight strands in strand diversification from 8 to 22 strands at an E-value $\leq 10^{-4}$. 8-stranded proteins are found to align to proteins with 10, 12, 14 and 16 strands. The strands that initially align from the 8-stranded barrels in the 16-stranded barrels can be aligned from 16-stranded barrels to 18-stranded barrels. Moreover, the strands that align between the 8- and 14-stranded barrels align between the 14- and 22-stranded barrels. However, 8-stranded barrels themselves do not directly align with the 18- or 22- stranded barrels. Arrows indicate alignment between proteins of different strand numbers. Pink strands align with pink strands, red strands align with red strands, etc. An asterisk indicates multiple patterns are observed as described in the text though only the most prevalent one is shown. White strands are strands that are not found to align to the original eight strands between barrels of different sizes, although they may be part of the alignment between proteins of different strand numbers. Examples of sequence alignments between barrels of different sizes and the lengths of the alignments are in supplement *Figure 4—figure supplement 1*, *Figure 4—source data 1* respectively.

DOI: https://doi.org/10.7554/eLife.40308.011

The following source data and figure supplement are available for figure 4:

**Source data 1.** Longest sequence alignments between barrels of different strand numbers.
DOI: https://doi.org/10.7554/eLife.40308.012

**Figure supplement 1.** Number of strands involved in alignments.
DOI: https://doi.org/10.7554/eLife.40308.013

observed in step one (32/33 cases representing relationships among 14,198 sequences included in the HMMs).

## Step 2

The diversification from 14 to 22 strands, and from 16 to 18 strands is the second step of the strand number diversification. The alignment from the 14- to the 22-stranded barrel is relatively homogeneous. All alignments are between strands 4–13 of the 14-stranded barrels and strands 10–19 of the 22-stranded barrels (*Figure 4—figure supplement 1H*).

The alignments from the 16-stranded barrels to 18-stranded barrels show extensive variation (*Figure 4—figure supplement 1I*), falling into four broad categories. The largest category (46% of the alignments observed at E-value $<10^{-3}$, representing relationships among 7815 sequences included in the HMMs) are a C-terminal alignment of two to ten strands in length, similar to that observed in step 1 with the 8-stranded barrels. The second-largest category (32% of the alignments, representing relationships among 8635 sequences included in the HMMs) are an N-terminal alignment in which the first strands align, the second strands align, etc., with between two and 16 strands involved. In the third-largest group (14% of the alignments representing relationships among 5897 sequences included in the HMMs), strands 1–14 of the 16-stranded barrel align with strands 4–18 of the 18-stranded barrel. Finally, in the fourth and smallest group 24 alignments (9%) have strands which align with loops or loops which align with strands. We sub-categorize these loop-to-strand alignments in the fourth category as 4A) loop to hairpin, 4B) loop to hairpin with alternate alignment, and 4C) large rearrangement (*Figure 5A*). This fourth category, in which loops and strands align, is perhaps the most interesting as it illustrates a potentially new, non-duplication method of strand number diversification.

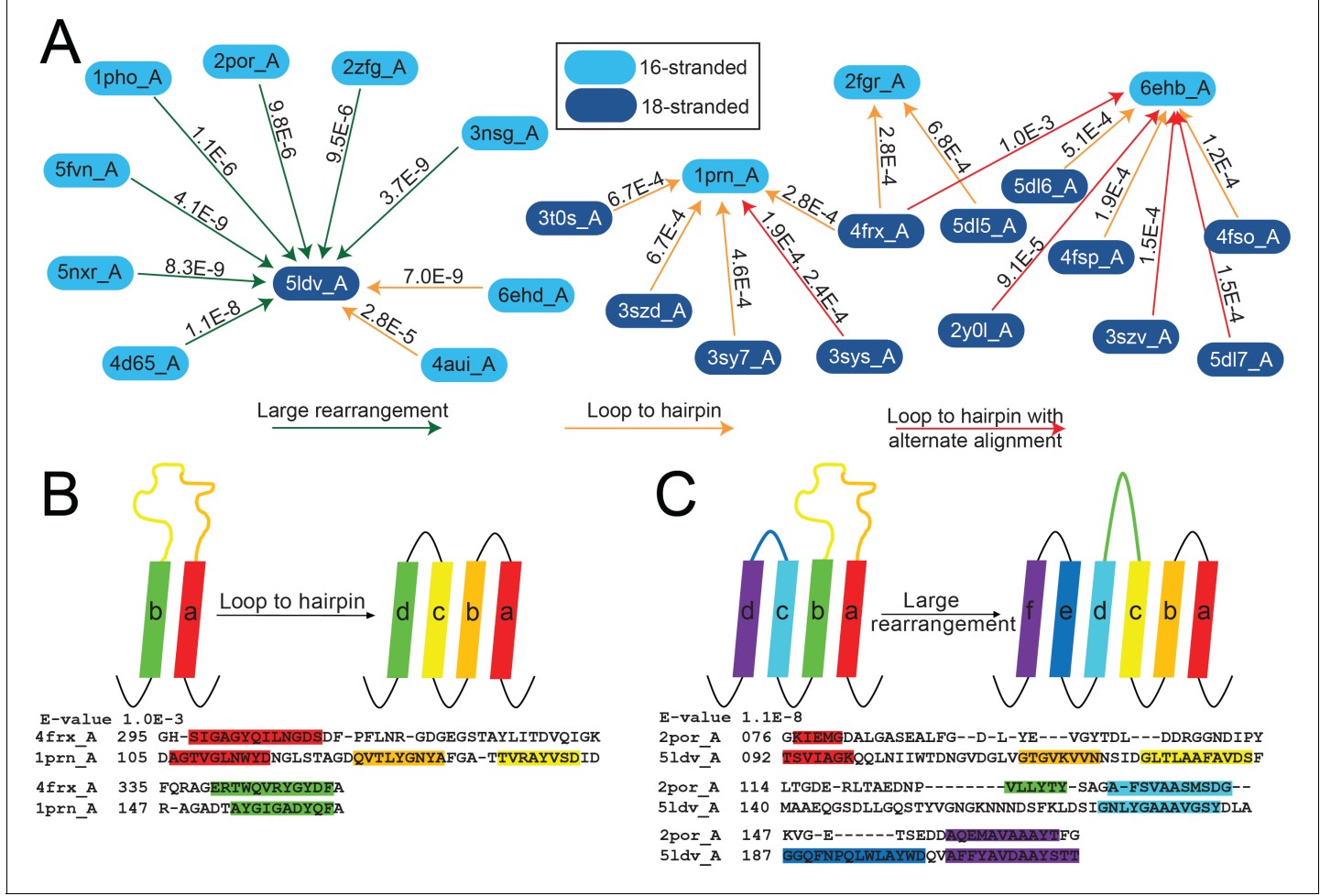

**Figure 5.** The unusual alignments of the 16- to 18-stranded barrels. (**A**) The subnetwork of unusual alignments with E-values shown above the arrow. 16-stranded barrels are shown in cyan; 18-stranded barrels are shown in blue. Full alignments with E-values are shown in *Figure 5—source data 1*. In (**B**) and (**C**) loops are colored to reflect the strands that they were or have become. Strand boundaries are determined by a combination of φ and φ angles and hydrogen bonding patterns as described in the materials and methods section. A representative alignment is shown below each diagram. Only strands are colored in the alignments to match the colors in the diagrams. (**B**) Loop to hairpin. A large loop in the left barrel (yellow/orange) becomes two strands in the right barrel. (**C**) Large rearrangement. In this rearrangement, the loop between a and b in the 16-stranded barrel becomes two strands in the 18-stranded barrel and strand b of the 16-stranded barrel becomes a loop in the 18-stranded barrel while the loop between c and d becomes a strand. Some barrels have alternative structures in which some amino acids in the extracellular region of the barrel may be defined as loop where in other structures they are defined as strand or vice versa. Only one of the 1653 alignments in the $10^{-3}$ E-value network changes strand number alignment based on alternative structure. That alignment is shown in *Figure 5—figure supplement 1* and does not change any data shown in *Figure 5*.

DOI: https://doi.org/10.7554/eLife.40308.014

The following source data and figure supplement are available for figure 5:

**Source data 1.** Full alignments of *Figure 5*.
DOI: https://doi.org/10.7554/eLife.40308.015
**Figure supplement 1.** Changed alignment using alternative structures.
DOI: https://doi.org/10.7554/eLife.40308.016

In order to trust these new but infrequent methods of strand number diversification, it is important that homologues of different strand numbers aren't heavily populating each other's HMMs leading profiles to have mistaken alignments between strands and loops. To confirm we haven't corrupted the strand identities in this way, we checked for homologues which overlapped between HMMs of different strand numbers. Only minimal overlap was found with the average percent

overlap between aligned sequences in barrels of different sizes at 0.041% and a maximum percent overlap of 3.7%.

The first of the three loop-to-strand alignment mechanisms is a loop to hairpin transition (*Figure 5B*) in which a long loop folds down to form a new hairpin between two existing strands. We observe two examples of this type of transition (*Figure 5A* orange arrows from cyan to blue) in which the 16-stranded loop folds down to form a hairpin in the 18-stranded barrel (representing relationships among 2903 sequences included in the HMMs). We also observe loop to hairpin transitions in which the 18-stranded barrel contains the loop and the 16-stranded barrel contains the hairpin (*Figure 5A* orange arrows from blue to cyan blue). We identify nine examples of this transition, representing relationships among 4398 sequences included in the HMMs. If this diversification occurred from the 16-stranded barrel to the 18-stranded barrel these represent a hairpin to loop transition. If this diversification occurred in the 18- to 16-stranded barrel direction, then it must be accompanied by a four strand deletion.

Additionally, there are six alignments (representing relationships among 3737 sequences included in the HMMs) that appear to be a loop-to-hairpin transition from 18 to 16 strands, but that also have a second alternative alignment in the same region (five red arrows in *Figure 5A*). Like internal repeats, alternative alignments are a hallmark of duplication. Of the six alignments, two have a second alternate loop-to-hairpin alignment while the remaining four loop-to-hairpin alignments have alternate C-terminal contiguous alignments.

The final type of loop-to-strand alignment is a large rearrangement in which there is a loop-to-hairpin transition, a strand to a loop transition, and a loop to a strand transition (*Figure 5A* green arrows and *Figure 5C*). Altogether that makes three loops and two strands in the 16-stranded barrels that are rearranged to form five loops and four strands. This has the unusual effect of aligning an extracellular-pointing (C terminus out) strand to a periplasmic-pointing (C terminus in) strand. We observe seven instances of this type of rearrangement, representing relationships among 3849 sequences included in the HMMs. These seven examples involve seven different 16-stranded proteins aligning to create the same large rearrangement with the same 18-stranded barrel (PDB ID 5ldv, *Figure 5A*). This is the same 18-stranded barrel involved in an alignment of a loop-to-hairpin transition as described above (*Figure 5A*).

Not only does the large rearrangement increase the total number of strands, it also results in a redirection of strand topology. Outside of the 16- to 18-stranded transitions, we find just four other examples of misdirection in strand topology, identified between four 14-stranded barrels (PDB IDs 3bs0, 3bry, 3dwo, and 1t16) and the same 16-stranded barrel (PDB ID 4y25), representing relationships among 3165 sequences included in the HMMs. In all four examples of 14- to 16-stranded barrels that have a topology change, strands 3–12 align to strands 4–16; an example of this alignment is shown in *Figure 4—figure supplement 1*.

In each of these loop-to-strand alignments there is generally more than one protein which aligns in the same way to the second protein. In these cases, it is unlikely that all proteins independently made the same loop-to-strand transition. More likely is that when one barrel evolved to another barrel, homologues of the original barrel maintained substantial similarity with the newly formed barrel or homologues to the descendent barrel arose that maintained homology with the original barrel. This similarity of original barrels can be seen in the seven 16-stranded barrels which align through a loop-to-hairpin transition to the 18-stranded barrel 5ldv (*Figure 5A*, left). The 16-stranded barrels are highly similar to each other (average E-value calculated in log space = $2.87 \times 10^{-23}$). Thus, it is likely that only one of these diversification events actually occurred and the others are a residual alignment to other closely related barrels.

What makes the loop-to-strand transition surprising is water-soluble loop regions becoming membrane-spanning β-strands and vice versa. This suggests that there is a low barrier to such a transition. We therefore tested for differences between loops and strands by assessing the polarity alternation and hydrophobicity of the aligned regions (see Materials and methods).

The alignments between 16-stranded barrels and 18-stranded barrels which contain large rearrangements tend to have a lower E-value than those that contain loop to hairpin transitions. However, it is unclear if the lower E-values are a feature of large rearrangements or a feature of aligning to the protein 5ldv. We note that 5ldv is the only trimeric, prototypical 18-stranded barrel and that all the 16-stranded barrels it aligns to that have a large rearrangement are also trimers. Due to the small numbers of examples of these transitions we cannot conclusively determine the significance or

origin of the lower E-values associated with the large rearrangements between 5ldv and the 16-stranded barrels.

Beta-strands have a sequence hallmark of alternating polar and non-polar amino acids. Therefore, we checked for an increase in polarity alternation between the residues which are part of the alignments that are loops in one structure and strands in another structure. The average polarity alternation in the loop conformation is 45% and the average polarity alternation in the strand conformation is 55%. However, both polarity alternation percentages are more like the alternation percentage of our control set of strands (44%) than the alternation percentage of our control loops (28%).

In contrast to the similarity in polarity alternation between loops and strands, we found that the loop regions were less hydrophobic than the strand regions with which they aligned although they are more hydrophobic than the loops in our control. We averaged hydropathy values (*Kyte and Doolittle, 1982*) for the amino acids which are part of the alignments and are loops in one structure and strands in another structure. The average hydropathy in the loop conformation is −0.57 and the average hydropathy in the strand conformation is −0.22 (negative is hydrophilic and positive is hydrophobic). Though these differ from each other, they are more hydrophobic and less different than our control loops and control strands which we found to have hydropathy scores of −1.11 and −0.40 respectively.

## C-terminal conservation

Determining the positional preference of alignments between barrels of different strand numbers allows us to trace a conserved, possibly 'original', set of strands through all the barrel sizes (*Figure 4*). The last seven strands of the 8-stranded barrels can be traced to the last strands of the 10-, 12-, 14-, 16- barrels, and then to the last strands of the 18-stranded barrels and to the middle of the 22-stranded barrels.

Because many of the alignments that we find are between fragments of proteins, we can assess the alignment preference for fragments at different positions within the protein. For each strand-position of each barrel of the same strand number (all strand 1s of 8-stranded barrels, all strand 2 of 8-stranded barrels, etc.) we assessed the number of times that strand position participated in an alignment. Documenting these alignments led to the observation that we find a significant conservation (i.e., reuse of sequence between proteins) of strands in the C-terminal half of the protein and a much lower conservation overall for the N-terminus (*Figure 6*, *Figure 6—figure supplement 1* at 25% sequence identity shows a similar trend). We find this C-terminal half's conservation to be the case for proteins of all strand numbers.

## Discussion

Looking at a combination of structural and sequence information allows us to trace how genetic changes influence structural changes, revealing evolutionary pathways and challenging the relationship between genetic duplication and structural repetition of repeat proteins. For this analysis high resolution structures are necessary to find these detailed evolutionary pathways. The 97 experimentally solved structures of prototypical barrels in our data set map to 50,832 homologous sequences incorporated into our HMM alignments. Given that the total number of OMBB sequences was assessed to be 48,731 (*Freeman and Wimley, 2012*) or 76,760 (*Reddy and Saier, 2016*), our methods have good coverage of the space of the non-structurally characterized OMBBs.

## Prevalence of divergence from mutations vs. duplication

Our analysis of the relative frequency and quality of alignments between barrels of the same size and barrels of different sizes illustrates that alignments are always more likely to be among barrels of the same strand number than between barrels of different strand numbers (*Figure 2—source data 1*). Moreover, as demonstrated by lower E-values, the quality of the alignments between barrels of the same strand numbers are better than those between barrels of different strand numbers. This difference in alignment quality demonstrates that for OMBBs partial-gene duplication events that are fixed are rare, despite duplication events being more common than mutation events generally. Strand-number changes may thus be understood as discrete events which can be charted as steps of evolutionary diversification.

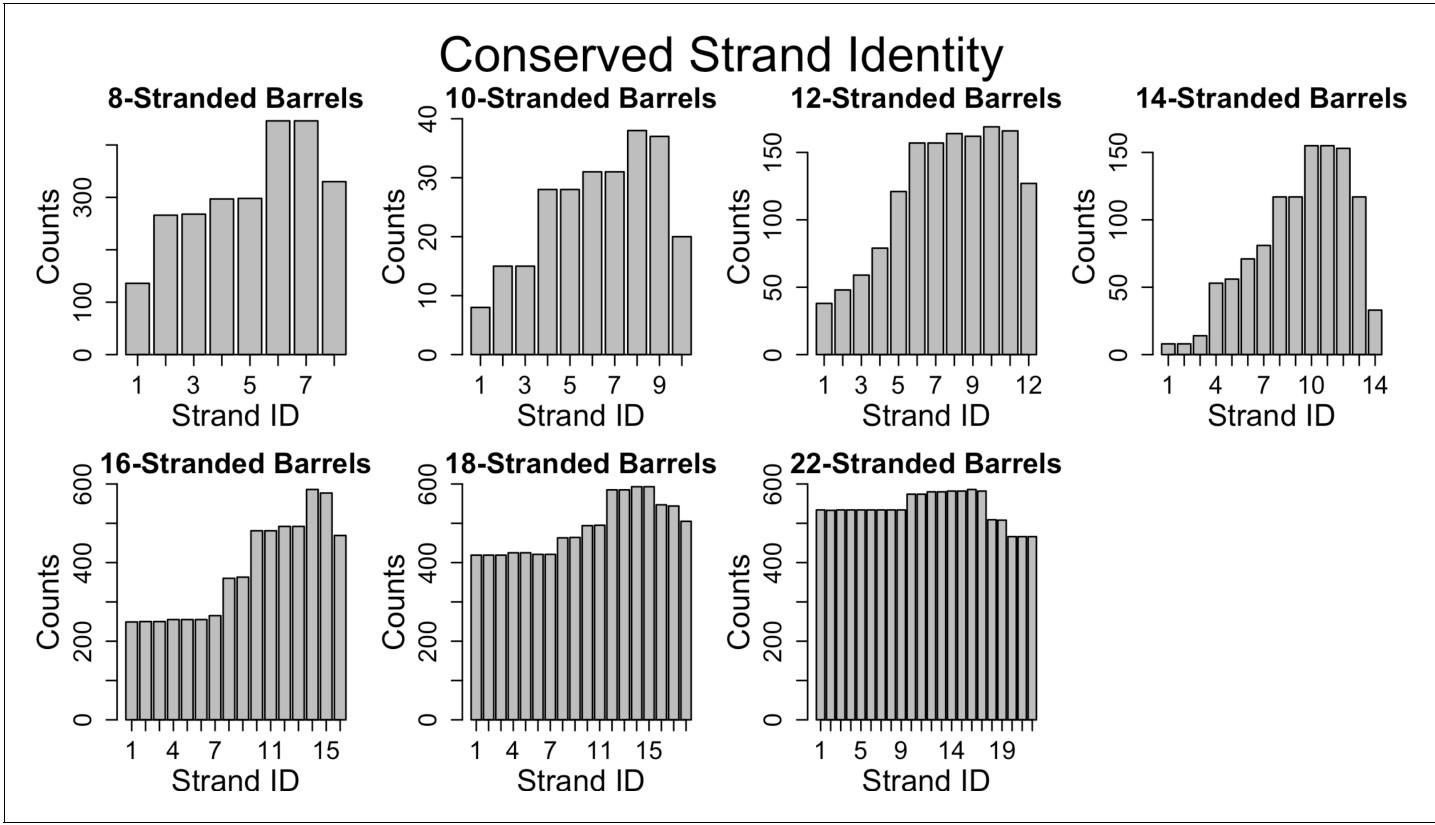

**Figure 6.** Conservation of strands in the prototypical barrels at <85% sequence identity. For each alignment involving two prototypical barrels, we determined the identity of the strands in each barrel. Each graph represents the distribution of the strands reused in that barrel size. *Figure 6—figure supplement 1* shows a similar trend at 25% sequence identity. The length distribution of these alignment (at 85% sequence) is shown in *Figure 6—figure supplement 2*.

DOI: https://doi.org/10.7554/eLife.40308.017

The following figure supplements are available for figure 6:

**Figure supplement 1.** Identity of stands in alignments at 25% sequence identity.

DOI: https://doi.org/10.7554/eLife.40308.018

**Figure supplement 2.** Number of strands in alignments.

DOI: https://doi.org/10.7554/eLife.40308.019

## Internal repeats and the primordial barrel

The large percentage of prototypical barrels that have internal alignments suggests that strand number diversification – when it occurs – largely results from amplification. Our data are consistent with these duplications occurring for many different numbers of strands. Although the 8-stranded barrel is the smallest known OMBB single chain barrel, we see an internal double hairpin shift, an interval of four strands from each other. Strands 1–3 align with strands 5–7 in 8/14 of 8-stranded barrels. This alignment points to the idea that the 8-stranded beta barrel evolved from a smaller segment, likely the double hairpin described previously (*Remmert et al., 2010*). Given the large number of 3- and 7-stranded alignments (*Figure 6—figure supplement 2*), the structure further implies that the last strand of these double hairpin units has degenerated, possibly due to evolutionary pressure.

A primordial 8-stranded barrel with an internal double hairpin repeat raises the possibility that a primordial four strands formed a multi-chain beta barrel. These strands would then have been duplicated to form the first single-chain barrel. Since most repeat proteins do not have function as individual domains, it has been hypothesized that many repeat proteins were originally homo-oligomers (*Ponting and Russell, 2000*). This hypothesis supports the idea of the primordial barrel as a multi-chain beta barrel like YadA (*Franklin et al., 2018a*): a trimeric adhesion protein, where each chain contributes four strands (a double hairpin) to create a 12-stranded barrel.

## A pathway for strand number diversification

Our graph of strand number connectivity (*Figure 4*) demonstrates a likely evolutionary path towards strand number diversification. The alignment data are consistent with the notion of a primordial 8-stranded barrel that developed more strands at the N-terminus through hairpin duplication (or some other evolutionary mechanism) to ultimately create 10-, 12-, 14-, and 16-stranded barrels. From these branching points, there were two secondary evolutionary transitions: 22-stranded barrels evolved from the 14-stranded barrels and 18-stranded barrels evolved from the 16-stranded barrels.

For many of the branch points there is only one location of the aligned strands from one strand number to another. The consistency of that location indicates that either there was a single event that caused the additional strands and then diversification from that point, or that there is a preferred pathway of duplication which all strand-increase events follow. The fact that the 18- and 22-stranded barrels overwhelmingly have alignments among them that encompass the entirety of both barrels (*Figure 6* and *Figure 6—figure supplement 2*) suggests that many barrels of these sizes are orthologs.

## Novel diversification mechanisms

The pattern of conservation in the 16- to 18-stranded barrels illustrates that a repeating topology can be created without motif duplications. Moreover, the alignments between 16- and 18-stranded barrels suggest more than one evolutionary path to an addition of two strands. Since we are hypothesizing that the 16-strand to 18-strand barrel transition event is among the more recent of the strand number diversification events (i.e., step 2), it makes sense that we can map these events more clearly than the other strand diversification events. These novel diversification mechanisms may have been used by other strand number transitions, but as those transitions would be older they would in turn be more likely to have been obscured by further evolution.

Some of the transitions between 16- and 18-stranded barrels occur through large rearrangements and shifts that used existing amino acids to generate the addition of two new strands rather than generating them through partial gene duplication (*Figure 5*). The types of rearrangements we see imply that the boundary between strand and loop may be somewhat fluid. Since half of strand residues are hydrophilic, membrane-bound strands are not too hydrophobic to become loops. Conversely, loops can become strands. In comparing the differences in hydrophobicity and in polarity alternation between the loops that become strands we find that for both qualities the loops and strands aligned in the transitions are more similar to each other than other loops and strands are similar to each other. It may be that the accretion of a few mutations from polar to charged amino acids cause a tight turn to be more favorable and insertion into the membrane less favorable. Alternately, we cannot rule out the possibility that the loop to strand transitions and the large rearrangements are the result of indels. We hope that further research will bring more clarity to these strand number diversification mechanisms.

The novel mechanisms of strand number diversification are a strong counterargument to the intuitive understanding that all repeat segments in repeat proteins are made from gene/motif duplications. Moreover, the creation of a repeat segment by something other than a duplication event indicates that both the environment and function of OMBBs required the repeated structural motif. In the process of collecting this data, we created other iterations of the alignment dataset, with slightly fewer sequence inputs. These iterations generated different alignments, but presented the same novel mechanisms of diversification with similar rates of occurrence. Therefore, it appears that the environment and function of the barrel enforces the structural repeats of hairpins in the beta barrel, even without sequence repeats.

## Transition from 14- to 22-stranded barrels

The mechanism of amplification for the 14- to 22-stranded barrel transition remains opaque. It may be that barrels larger than 18 strands require a large stabilizing domain plugging the pore, such as the β-sandwich observed in the middle of the 22-stranded barrels. The 14-stranded barrels from which the 22-stranded barrels evolved have small N-terminal plug domains with tertiary structure; other barrels with more than 10 strands have occluding loops but not domains with tertiary structure. The presence of the more structured plug could have facilitated the 22-stranded barrels to branch off of the 14-stranded barrels. However, the unusual, non-C-terminal localization of the

strands from the 14-stranded barrel in the 22-stranded barrel, as well as the overall lack of internal duplication in 22-stranded barrels, implies that this transition used a different style of amplification than the mechanisms described here. We hope that future analysis will explain this puzzling transition.

## No change in direction

The existence of strand number diversification events that are not hairpin duplications means that strand number diversification could occur with single strand additions. This raises the question: if single strand additions are possible, why are all bacterial OMBBs even-stranded? We believe that this is because the strands and loops encode a membrane topology that is difficult to override. Unlike inner membrane proteins with a positive inside rule (*von Heijne and Gavel, 1988*), outer membrane proteins show a charge-out rule within their strands (*Seshadri et al., 1998*; *Slusky and Dunbrack, 2013*). Addition of an odd number of strands anywhere but the C-terminus would change the directionality of all subsequent strands. The preference for directional maintenance is so intense that of all 2642 alignments represented in *Figure 1*, there are only 11 instances of strand redirection (*Figure 5A and C*, *Figure 5—source data 1*) in which any strands change topology (from N-terminus periplasmic to N-terminus extracellular or vice versa). The 11 instances we document (*Figure 5A*, *Figure 5—source data 1*) representing relationships among 6978 sequences in the HMM profiles) all have a single strand that flips from the N-terminal end of the strand facing into the cell to the N-terminal end of the strand facing out of the cell. However, in all cases the topology flip is then corrected by a subsequent strand becoming a loop so that the topology reversal is not perpetuated through the entire protein. The topological rigidity of duplications in beta barrels contrasts with duplications in helical proteins, which are distributed roughly evenly between parallel and antiparallel duplications (*Hennerdal et al., 2010*).

## Increasing strand numbers more prevalent than decreasing

The relational network shows frequent strand additions; however, it is less clear on the prevalence of strand subtractions. If *Figure 1* is viewed with a primordial 8-stranded barrel (*Figure 4*) in mind, all nodes of a lower strand number branch off groups of nodes of a higher strand number. The most ambiguous example is the unusual case of 12-stranded 2wjr, which aligns to 8-stranded barrels and 16-stranded barrels. Since many of the 8- and 16-stranded barrels align with each other, it is unclear if this 12-stranded barrel evolved from the gain of four strands from the 8-stranded barrels or if it is an unusual case of a loss of four strands from the 16-stranded barrel.

Though we find no clear cases of strand deletion, we cannot exclude the possibility that some of the alignments between larger and smaller barrels are deletions from the large to the small barrel rather than duplications from the small to the large barrel. For example, some of the alignments between 16- and 18-stranded barrels may originate at the 18-stranded barrel and result in the 16-stranded barrel. The observed loop-to-hairpin transitions from 18-stranded barrels to 16-stranded barrels, although not evidence of a deletion, may be evidence of a mechanism of diversification that can proceed from larger barrels to smaller barrels. We attribute the lack of any unambiguous strand loss events to any one of the following three causes, or a combination thereof: 1) The rate of gene or domain deletions in bacteria (*Nilsson et al., 2005*) is generally regarded to be lower than that of gene or domain duplications in bacteria (*Starlinger, 1977*; *Reams et al., 2010*; *Katju and Bergthorsson, 2013*); 2) Because the primordial, 8-stranded, single-chain barrel appears to be the smallest, starting from that barrel would require two strand change events—one addition and then one subtraction—in order to see a subtraction; 3) Strand change events are sufficiently difficult to fix and happen infrequently enough that the sample size is underpowered to detect them.

## No 20-stranded barrels

It is surprising that structures have been resolved for every even number of stranded barrel from 8 to 26, with the exception of 20-stranded barrels (*Figure 3A*). Even the more permissive search used by Reddy and Saier, which predicts over 50,000 barrels, does not find any 20-stranded OMBBs for bacteria (*Reddy and Saier, 2016*).

Our sequence similarity networks shed some light on this mystery. If 20-stranded barrels require a large plug for stabilization, they would need to have diversified from the small set of large-plug-

containing barrels, as the 22-stranded barrels did. Moreover, OMBB strand numbers are clearly not randomly distributed. Our network demonstrates that strand number transitions are rare, likely requiring distinct and improbable events. Based on the type of transitions we have seen, the most likely pathways would be either a doubling of a 10-stranded barrel or one of the mechanisms of strand addition seen in the 16-to-18 strand transition, applied to an 18-stranded barrel. However, 10-stranded barrels are rare enough themselves, with only three occurrences in our data set. More-over, a strand addition from 18 would be three transitions away from the primordial group, a num-ber of steps that has not yet been documented. Given the rarity of amplification events, it may be that 20-stranded barrels have not yet been sampled.

## C-terminal strand conservation

While prior work has shown that membrane proteins diversify more than other proteins (*Shimizu et al., 2004*; *Sojo et al., 2016*), we extend these results to demonstrate that this diversifi-cation is applied unequally. It is easier to find evolutionary traces in the C-terminal half of outer membrane proteins than in the N-terminal half. This is an overarching feature of OMBBs; however, it is most distinct for 16-stranded barrels (*Figure 6*, *Figure 6—figure supplement 1*). These results are either due to negative selection (i.e., suppression of fixation in the C-terminal half), or positive selec-tion that leads to diversification in the N-terminal half.

The importance of the penultimate strands of the OMBBs has been observed before. For exam-ple, mutational studies of 8-stranded OmpA identify strand six as being particularly important for folding (*Koebnik, 1999*; *Stapleton et al., 2015*). Moreover, constructs containing duplicated strands of the 8-stranded protein OmpX also successfully folded, as long as strands 4 and 5 were included (*Arnold et al., 2007*). The prominence of the penultimate strands in OMBB folding may explain unsuccessful attempts at design as well. For instance, the unsuccessful attempt to increase the num-ber of strands in OmpG by duplicating the final hairpin pushed the penultimate hairpin out of the penultimate position(*Korkmaz et al., 2015*).

The conservation in the C-terminal half does not appear to be related to the beta signal sequence (*Struyvé et al., 1991*), which would be located in the final strand of the barrel as the final strand appears less conserved. Rather, this pattern of reuse may be a result of (post-translationally) folding the C-terminal half of the barrel first, as has recently been posited (*Maurya and Mahalakshmi, 2016*). If this is shown to be true for other proteins, it may suggest that C-terminal conservation pro-tects the folding pathway. Previous investigation of soluble proteins has shown that the folding nucleus of proteins is more conserved than the rest of the protein, in order to preserve the ability of the protein to fold (*Mirny and Shakhnovich, 2001*). The C-terminal half may nucleate folding by inserting the first beta-strands, thereby creating a template for further beta-strand formation (*Figure 7A*). Alternately, this C-terminal half conservation supports the folding mechanism might be similar to the mechanism recently proposed for the mitochondrial SAM50 (*Höhr et al., 2018*). In this scenario, BamA would incorporate C-terminal strands of the unfolded client protein into the assem-bly barrel with the N-terminus of the client protein held in the pore of the assembly protein (*Figure 7B*). Future research must assess the folding mechanism of more bacterial, prototypical beta barrels before the reason for conservation of the strands on the C-terminal half of the protein can be fully understood.

Overall, mapping sequence alignments with solved structures of prototypical beta barrels sheds light on the evolution of these diverse proteins and illustrates new mechanisms of how genetic changes influence structural changes. As greater numbers of prototypical beta barrels are structur-ally characterized, more links will be elucidated. However, even with many more structurally resolved beta barrels, we may never see the branches that fully map the diversification space as steps are swallowed by evolution.

## Materials and methods

### Barrel network creation

The barrel network was created following the method previously described (*Franklin et al., 2018a*). Briefly, we generated an HMM profile for each barrel in our set of 138 proteins from the database uniclust30_2017_10 (available at http://wwwuser.gwdg.de/~compbiol/uniclust/2017_10/) using

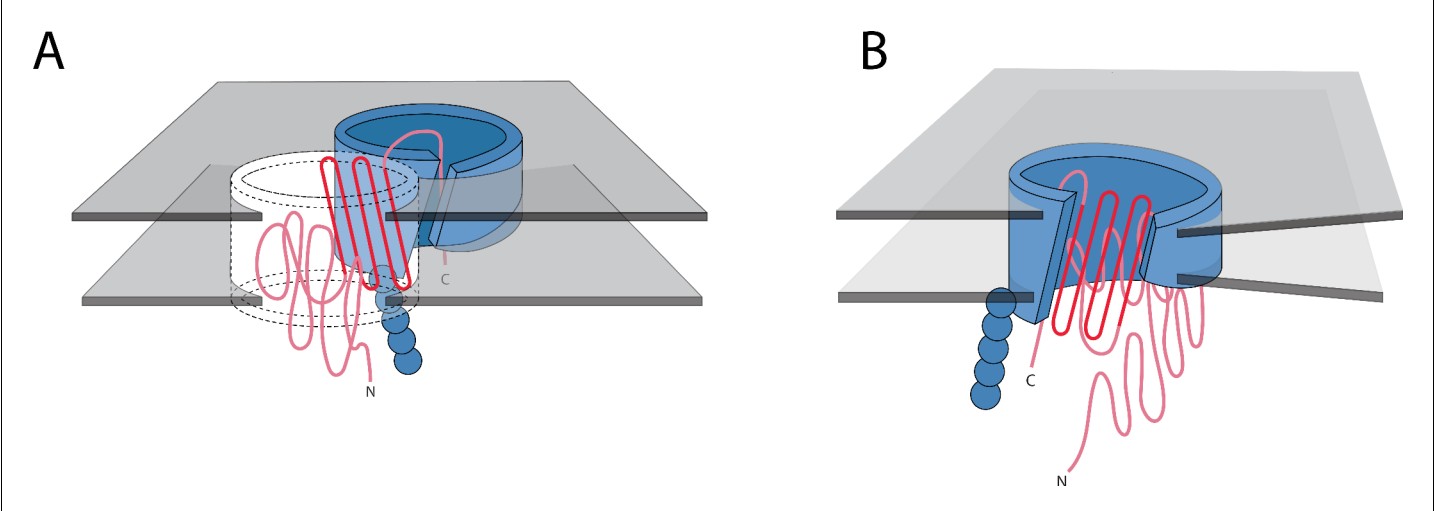

**Figure 7.** Possible mechanisms of OMBB folding. The C-terminal half of beta barrels evince a strong evolutionary trace. The strong evolutionary trace may be indicative of nucleation. (**A**) Nucleation on side of barrel: Such nucleation could occur with the second half of the nascent barrel (red squiggly lines) forming beta hairpins in the membrane (boundaries shown with gray planes) along strands of the outer membrane translocation machinery, BamA with POTRA domains (solid blue barrel with five attached spheres). The rest of the nascent barrel would use the formed strands as a template. (**B**) Nucleation in incorporation of barrel: the strong evolutionary trace may also be indicative of beta strand incorporation, whereby the second half of the beta barrel is incorporated into BamA. In this model, the conserved strands form new strands in the BamA barrel while the N-terminal half of the protein inserts through the center of the barrel. Ultimately the nascent barrel buds off of the BamA, creating a new OMBB.
DOI: https://doi.org/10.7554/eLife.40308.020

default parameters for HHblits including a MAC threshold at 0.35. HMM-HMM aligners are more sensitive than using PSI-BLAST or HMMER3 for building iterative HMM profiles (*Remmert et al., 2011*). The MSA in a3m format for our HMMs are available online at https://github.com/SluskyLab/OMBB_A3Mfiles.git (copy archived at https://github.com/elifesciences-publications/OMBB_A3Mfiles). Then, we identified evolutionary relationships using the sensitive HMM sequence aligner HHsearch (*Söding, 2005*). HHsearch uses local alignments, meaning that it is not forced to align full sequences. Rather it can align subsections within proteins and can identify more than one possible alignment. We applied a 20-residue cutoff which is a length that includes 99.7% of hairpins in our database. The prototypical barrel cluster was defined as the large cluster that had no connection with an E-value $\leq 10^{-3}$ to any other OMBB.

We organized the set of proteins and their alignments as a network, using Cytoscape (*Saito et al., 2012*), and CyToStruct (*Nepomnyachiy et al., 2015*) to easily view the alignments in a molecular viewer. The resulting network is available online at: http://cytostruct.info/rachel/protos/index.html. Readers can change the E-value cutoff and see the resulting networks; for instance, using a more stringent cutoff removes edges, resulting in a more fragmented network. The structure of a protein can be viewed in a molecular viewer by clicking its node. The structural and sequence alignment can be viewed by clicking the edge.

Strands were defined using in-house software, Polar Bearal, as previously described (*Slusky and Dunbrack, 2013*; *Franklin et al., 2018a*), available from https://github.com/SluskyLab/PolarBearal.git (copy archived at https://github.com/elifesciences-publications/PolarBearal). This uses the phi/psi angles to label residues as helix, strand, or other secondary structure. The strands are defined by hydrogen bonding between regions of strand-like residues.

Some barrels have alternative structures in which some amino acids in the extracellular region of the barrel may be defined as loop where in other structures they are defined as strand or vice versa. Barrels for which multiple structures are available, were analyzed using the longest and shortest strand definitions. Using alternative structures alters the number of strands in only one of the 1653 alignments in the $10^{-3}$ E-value network (*Figure 5—figure supplement 1*). That alignment does not change the features reported here nor is it one of the transitions described in *Figure 5*.

## Internal repeats

Internal repeats were identified by self-aligning each of the 138 β-barrels using HHalign. Results were filtered to only include alignments with an HHalign probability >75% and E-value $\leq 10^{-3}$. The HHalign probability is the log-odds probability of whether the alignment was produced by the HMM versus a random model (*Söding, 2005*). The full set of results can be found in the supplemental file InternalRptsE10-3.xlsx.

## Polarity alternation and hydrophobicity calculations

Alignments of 16-and 18-stranded β-barrels where strands aligned with loops were selected for hydrophobicity and alternating polarity assessment. Any residue that aligned with a gap was not used in the calculation. Strand/loop regions that had alignments with multiple other strand/loops were included if they contained less than 50% amino acid identity compared to the same region in other alignments. Control datasets for both loops and strands were created using all the strands and extracellular loops in 16- and 18- stranded β-barrels that were not part of any loop-strand transition alignment.

Hydrophobicity was calculated as the average of each residues' hydrophobicity (*Kyte and Doolittle, 1982*) in that strand/loop region. Alternating hydrophobicity was calculated by dividing the number of residues with opposite polarity of the previous amino acid by the total number of residues. Hydrophobicity alternation was calculated for residues after the starting residue of the strand/loop and did not include residues immediately after a sequence gap. A,F,I,L,M,P,V,W and Y were considered nonpolar. C,D,E,G,H,K,N,Q,R,S, and T were considered polar.

## Acknowledgements

Professors James Walters, Mark Holder, Christian Ray, Eric Deeds, and Andrei Lupas for helpful discussions, Amy Weiss and Gil Ortiz for illustrations and Tara Mendola for edits. NIH awards DP2GM128201, P20GM103418, P20GM113117, T32-GM008359, NSF MCB160205, The Gordon and Betty Moore Inventor Fellowship, KU-startup, and Israel Science Foundation grant number 450/16.

## Additional information

### Competing interests

Nir Ben-Tal: Reviewing editor, *eLife*. The other authors declare that no competing interests exist.

### Funding

| Funder | Grant reference number | Author |
| --- | --- | --- |
| National Institute of General Medical Sciences | DP2GM128201 | Meghan Whitney Franklin Joanna SG Slusky |
| Gordon and Betty Moore Foundation | Moore Inventor Fellowship | Joanna SG Slusky |
| National Science Foundation | MCB160205 | Joanna SG Slusky |
| Israel Science Foundation | 450/16 | Nir Ben-Tal Rachel Kolodny |
| National Institute of General Medical Sciences | P20GM103418 | Meghan Whitney Franklin Joanna SG Slusky |
| National Institute of General Medical Sciences | P20GM113117 | Meghan Whitney Franklin Joanna SG Slusky |
| National Institute of General Medical Sciences | T32-GM008359 | Meghan Whitney Franklin Joanna SG Slusky |

The funders had no role in study design, data collection and interpretation, or the decision to submit the work for publication.

## Author contributions
Meghan Whitney Franklin, Software, Investigation, Methodology, Writing—original draft, Writing—review and editing; Sergey Nepomnyachyi, Software, Methodology; Ryan Feehan, Software, Investigation, Methodology; Nir Ben-Tal, Rachel Kolodny, Software, Methodology, Writing—review and editing; Joanna SG Slusky, Conceptualization, Writing—original draft, Writing—review and editing

## Author ORCIDs
Nir Ben-Tal (iD) http://orcid.org/0000-0001-6901-832X
Rachel Kolodny (iD) http://orcid.org/0000-0001-8523-1614
Joanna SG Slusky (iD) http://orcid.org/0000-0003-0842-6340

## Decision letter and Author response
Decision letter https://doi.org/10.7554/eLife.40308.027
Author response https://doi.org/10.7554/eLife.40308.028

# Additional files

## Supplementary files
• Supplementary file 1 PDB list. List of PDBs in the prototypical group. The format is PDB_chainID.
DOI: https://doi.org/10.7554/eLife.40308.021

• Transparent reporting form
DOI: https://doi.org/10.7554/eLife.40308.023

## Data availability
All data generated is available on the website http://cytostruct.info/rachel/protos/index.html. Summary files of the results are included in the supplement. Software is available on GitHub https://github.com/SluskyLab/PolarBearal.git (copy archived at https://github.com/elifesciences-publications/PolarBearal) as are the a3m files https://github.com/SluskyLab/OMBB_A3Mfiles.git (copy archived at https://github.com/elifesciences-publications/OMBB_A3Mfiles).

The following previously published dataset was used:

| Author(s) | Year | Dataset title | Dataset URL | Database and Identifier |
|---|---|---|---|---|
| Johannes Söding | 2017 | uniclust30_2017_10 | http://wwwuser.gwdg.de/~compbiol/uniclust/2017_10/ | Uniclust, uniclust30_2017_10 |

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
