## [Decision Letter]

[Editors’ note: a previous version of this study was rejected after peer review, but the authors submitted for reconsideration. The first decision letter after peer review is shown below.]

Thank you for submitting your work entitled "Evolution of Environmentally-Enforced, Repeat Protein Topology in the Outer Membrane" for consideration by *eLife*. Your article has been reviewed by two peer reviewers, and the evaluation has been overseen by Vikas Nanda as Guest Editor and Detlef Weigel as Senior Editor. The following individuals involved in review of your submission have agreed to reveal their identity: Vikas Nanda (Reviewer #1).

Our decision has been reached after thorough consultation between the reviewers. The problem of membrane barrel evolution is a very important one that is challenging due to constraints of the membrane environment, repeat structure and low-complexity sequence on evolution; however, there were a number of concerns raised regarding the size of the data set and lack of statistical power to support the conclusions drawn. Based on these discussions and the individual reviews below, we regret to inform you that your work will not be considered further for publication in *eLife*.

Reviewer #1:

The emergence and evolutionary divergence of OMBBs is a fascinating problem due to the highly regular topology (tandem up-down β-hairpins, even number of strands). The authors combine sequence alignments with structural insight to propose a novel, compelling model for how OMBBs evolved. The evolution of repeat proteins is challenging as Elofsson has shown where the evolutionary and structural repeat unit do not necessarily coincide. In Slusky's study – the barrel evolution is more highly constrained than one might expect. A number of new and interesting observations are made in this study, such as the placement of 8-stranded barrels at the root of the OMBB tree, and all other barrels contain evolutionary traces of the original 8-stranded motif. Despite strand amplification being the most compelling mechanism for OMBB diversification, it appears the fixation of duplications was rare. I thought the insight of loop-topology as the primary constraint on the ubiquitous even-strandedness to be quite insightful – this makes a lot of sense.

I found the proposed transitions between 16- to 18- stranded barrels (Figure 5) to be interesting, but difficult to imagine occurring through processive single mutations. A number of concerted mutations would have to occur to convert loops into strands and vice-versa. How would transitions in the 'large rearrangement' mechanism for example be viable? Might such transitions come from multi-residue indels, rather than successive single amino acid changes?

I was hoping for some answer to the mystery of the missing 20-stranded barrel Understandably, it is harder to explain absence evolutionarily. It seems the same processes that lead 16 to 18, could also lead 18 to 20. Likewise, if 8 can go to 14, 14 can go to 20. If 8 can go to 12, 16 can go to 20. >3 amplification steps are not required with the proposed evolutionary model. Perhaps the answer lies in the rarity of amplification events overall and 20 has not been sampled.

Reviewer #2:

The authors study OMBB path of diversification by tracing sequence alignments among their solved structures and finds some intriguing patterns which seems to fit in a story about the evolutionary mechanism of these proteins. Although the story looks plausible, some evidences and statistic tests should have been done to support it.

1) Because the study focuses on structurally resolved proteins, this creates a bias towards sequences/families that have been studied structurally. There is also a bigger issue about the number of structurally resolved OMBBs, 130 structures are too less for an evolutionary study. Most evolutionary studies start with several thousand sequences.

2) The author's claims highly depend on the similarity between proteins/strands, which highly depends on the cut-off E-value of the alignments. This cut-off determines how significant the found patterns are. Since the paper says most β-barrel proteins are homologous (Introduction), I would expect a very low E-value cut-off should have been used in the study. Instead, the paper uses some relatively large values (10^-1^~10^-3^). Certain statistic tests or at least some explanations are required to justify the usage of such E-value cut-off.

3) The OMBBs with 22 strands do not seem to be evolutionary related to other OMBBs at lower e-values. Even at e-value of 10^-2^, a large number of 22 OMBBs do not align with 14-stranded OMBBs. How can the 22 strands BB evolve so fast that they no longer can align with 14 strand OMBBs? Supposedly the 14 to 22 strand evolutionary event is a relatively new one.

4) The paper finds some interesting patterns that connect β-barrel proteins with different sizes together. However, all the results shown look like case study, and few statistics are reported to show how prevalent and significant these patterns are in this protein population.

Here I list only one example to show this issue. In subsection “Internal repeats”: "The 12-, 14-, 18-, and 22-stranded barrels have single hairpin shifts. The 8- and 16- stranded barrels sometimes have double hairpin shifts and sometimes have single hairpin shifts." What are the percentages of the 12-, 14-, 18-, and 22-stranded barrels that have single hairpin shifts and what is the "sometimes" for 8- and 16-stranded barrels are not clear. Concrete numbers should be reported instead of descriptive language.

5) What mechanism was responsible for events that do not involve complete gene duplication? e.g. 14 to 22 strands. The authors should consider mapping the TM strands on to the exons and explore if exon duplication might be one of the mechanisms that achieves this?

6) The authors should comment if an event similar to that of converting an extracellular facing strand to a periplasmic-facing strand has been observed in α-helical membrane proteins.

7) The writing needs a lot of improvement. The English of the paper is neither plain nor scientific. Just one example (subsection “Step 1”) to show this:

This manifests as "at least some" of the 8 strands of the 8-stranded barrels "almost always" (150/151 cases) aligning with "at least some" of last eight strands of the 10-, 12-, and 16-stranded barrels such that strand 1-"when it aligns"-aligns to the 8th to the last strand, and strand 2 to the 7th to last strand, 3 to the 6th to last strand, "etc.".

A sentence in a scientific paper should not spans four rows accompanied with a lot of vague terms (shown in quotes above) but lacking concrete statistic numbers, can make any point clear. And this is not an isolated case.

[Editors’ note: what now follows is the decision letter after the authors submitted for further consideration.]

Thank you for submitting your article "Evolutionary Pathways of Repeat Protein Topology in Bacterial Outer Membrane Proteins" for consideration by *eLife*. Your article has been reviewed by Detlef Weigel as the Senior Editor, a Reviewing Editor, and three reviewers.. The following individual involved in review of your submission has agreed to reveal his identity: Vikas Nanda (Reviewer #1).

The reviewers have discussed the reviews with one another and the Reviewing Editor has drafted this decision to help you prepare a revised submission.

Summary:

It is appreciated that large, complex proteins likely evolved through duplication or fusion of smaller ancestral domains. In the case of OMP β-barrels, the exclusive repeat of a β-hairpin like structure suggests that modern multi-stranded barrels emerged through a series of hairpin repeats. This work carefully examines that idea using network analysis and state-of-the-art sequence analysis methods based on hidden Markov Models (HMM) of the families represented by known structures, in order to examine sequence relationships and the origins of structural expansion. To date, the understanding of β-barrel fold evolution has been based on the careful work of Remmert et al., (2010), which showed that the β-stranded hairpins represents the primordial evolutionary unit that expanded through duplication events. This work builds on this basic framework and provides evidence for additional, alternative pathways for evolution of hairpins, for example, by insertion of loops.

Central conclusions:

1) Greater similarity is observed between barrels of the same strand number versus those of different size, indicating that hairpin amplification events are rare relative to mutational changes.

2) Alternative mechanisms for strand amplification are identified including loop-hairpin transitions and large-scale strand rearrangements.

3) The progression from ancestral 8-strand barrels to larger barrels proceeded through a set of defined intermediates of increasing strand number.

4) The C-terminal region of OMPs are most conserved across barrels of different strand number, suggesting a conserved folding-nucleation domain that can be traced back to the original family of 8-stranded barrels.

The reviewers find the study to be timely, rigorous, thoughtful, and well written and presented.

Essential revisions:

There was discussion among the reviewers regarding the extent to which including predicted secondary structure information would improve the analysis. Reports of secondary structure prediction as high as 87% have been reported (Ou et al., 2010). It’s unclear to what extent the assignments of evolutionary relationships would be sensitive to precise strand boundary definitions. We would like the authors to consider this question and justify the choice of eliminating sequences where structural data are not available.

Other points to be addressed:

1) "E-values are always lower among barrels of the same strand number than between barrels of a different strand number." Can the authors be sure that this is due purely to similarity, or is there a concern that either the cost of inserting gaps or the greater number of expected alignments adding to noise when using a shorter query to probe a longer target? This may be accounted for in the analysis, but it was not obvious how it was handled. Related to this, is it possible whether the lack of connections from the largest barrels (18- and 22-stranded) to the smallest (8, 10, 12-stranded), might reflect challenges to the scoring schemes when such length differentials are present. How sensitive are the results for these particular connections to lowering of the thresholds? A comment as to the reasons for choosing E <= 10^-3^ for selection of the cases in the prototypical barrel group (Results section), would be helpful.

2) The conservation of 8 strands throughout the barrel families (depicted in Figure 4) is fascinating, particularly because one would expect that larger barrels would require larger amino acids to enforce the curvature of the barrel circumference. Since the authors know the register of the HMM profiles with respect to structure, can it be said whether this conservation is due to core vs surface residues? Either way – it would inform the discussion regarding the mechanism of folding. A conserved folding nucleus would imply core facing residues are conserved. A conserved surface would imply a different transition state in the presence of the BamA machinery. Of course, both could be true.

3) The authors are using 3D structure of the proteins to gather the strand start and strand register values. It is not clear to how the authors address the artifacts introduced in the X-ray structures. For example, crystals for BamA from *Salmonella* and *Haemophilus ducreyi* have been resolved (5ORI and 4K3C respectively). Strands 1 and 16 in both structures have significantly different lengths (3 and 5 in 5ORI and 8 and 10 in 4K3C). How is this resolved?

4) In "Transition from 14- to 22- stranded barrels", the authors claim the 18- stranded barrels do not have plugs. This is incorrect, most barrels > 10 strands have in-plugs.

5) Why are the barrels > 22 strands not included in the study (PapC, 2vqi; FimD, 3rfz; LptD, 4q35)?

6) For the average polarity and hydrophobicity of the loops and strands, it would be useful to have a control in a different region of the protein, as well as standard deviations to get a sense of the variability of these values in context.

7) It would be best if the in-house software Polar Bearal (subsection “Polarity alternation and hydrophobicity calculations”) were made available through GitHub or a similar portal.

---

## [Author Response]

[Editors’ note: the author responses to the first round of peer review follow.][…] Reviewer #1:The emergence and evolutionary divergence of OMBBs is a fascinating problem due to the highly regular topology (tandem up-down β-hairpins, even number of strands). The authors combine sequence alignments with structural insight to propose a novel, compelling model for how OMBBs evolved. The evolution of repeat proteins is challenging as Elofsson has shown where the evolutionary and structural repeat unit do not necessarily coincide. In Slusky's study – the barrel evolution is more highly constrained than one might expect. A number of new and interesting observations are made in this study, such as the placement of 8-stranded barrels at the root of the OMBB tree, and all other barrels contain evolutionary traces of the original 8-stranded motif. Despite strand amplification being the most compelling mechanism for OMBB diversification, it appears the fixation of duplications was rare. I thought the insight of loop-topology as the primary constraint on the ubiquitous even-strandedness to be quite insightful – this makes a lot of sense.I found the proposed transitions between 16- to 18- stranded barrels (Figure 5) to be interesting, but difficult to imagine occurring through processive single mutations. A number of concerted mutations would have to occur to convert loops into strands and vice-versa. How would transitions in the 'large rearrangement' mechanism for example be viable? Might such transitions come from multi-residue indels, rather than successive single amino acid changes?

This is an interesting possibility. In the previous calculations we found no significant difference in the alternation of polarity of the amino acids in strands and loops. In the revised manuscript we add an assessment of the average hydrophobicity of the regions that are strands in one alignment and loops in another:

“In contrast to the similarity in polarity alternation between loops and strands, we found that the loop regions were more hydrophobic than the strand regions with which they aligned. We averaged hydropathy values (Kyte and Doolittle, 1982) for the amino acids which are part of the alignments and are loops in one structure and strands in another structure. The average hydropathy in the loop conformation is -0.66 and the average hydrophobicity in the strand conformation is 0.01. These groups are statistically different, with a Wilcoxon rank sum test p-value of 0.01.”

In the Discussion section we have added a paragraph that refers to the possibility that there is a fair amount of fluidity between what is a strand and what is a loop and that mutations from polar to charged residues may maintain alternating polarity while increasing hydrophobicity. However, we cannot rule out the possibility of indels:

“…The types of rearrangements we see imply that the boundary between strand and loop may be somewhat fluid. Since half of strand residues are hydrophilic, membrane-bound strands are not too hydrophobic to become loops. Conversely, loops can become strands. Though we find the alternation of polarity to be the same between these loop and strand regions, average hydrophobicity is higher for strands than loops. The accretion of a few mutations from polar to charged amino acids may cause a tight turn to be more favorable and insertion into the membrane less favorable. Alternately, we cannot rule out the possibility that the loop to strand transitions and the large rearrangements are result of indels. We hope that further research will bring more clarity to these strand number diversification mechanisms.”

I was hoping for some answer to the mystery of the missing 20-stranded barrel Understandably, it is harder to explain absence evolutionarily. It seems the same processes that lead 16 to 18, could also lead 18 to 20. Likewise, if 8 can go to 14, 14 can go to 20. If 8 can go to 12, 16 can go to 20. >3 amplification steps are not required with the proposed evolutionary model. Perhaps the answer lies in the rarity of amplification events overall and 20 has not been sampled.

We thoroughly agree with this assessment and have added the following statement:

“… Given the rarity of amplification events, it may be that 20-stranded barrels have not yet been sampled.”

Reviewer #2:[…] 1) Because the study focuses on structurally resolved proteins, this creates a bias towards sequences/families that have been studied structurally. There is also a bigger issue about the number of structurally resolved OMBBs, 130 structures are too less for an evolutionary study. Most evolutionary studies start with several thousand sequences.

Thank you for these helpful comments. We hope you find the manuscript has improved.

For all of the alignments discussed in this manuscript we used HMMs not proteins. The HMMs are statistical models that represent sequences of the protein and many of its homologs; we align the HMMs (rather than the bare sequences) so as to include these additional protein sequences in our analysis. Thus, although we used only 97 proteins these are actually representatives of a total of 50,832 sequences.

We regret that we didn’t make this more apparent in the initial manuscript and have thoroughly updated our manuscript to reflect this. For example, when describing the loop to hairpin transition we now write:

“We observe four examples of this type of transition in which the 16-stranded loop folds down to form a hairpin in the 18-stranded barrel (representing relationships among 3806 sequences included in the HMMs).”

With respect to the bias introduced by using the structurally solved dataset, we find that structure prediction methods are not correct/detailed enough to find the molecular evolution mechanisms that we suggest. We have added the following to the Introduction:

“Here we explore OMBB diversification through structural similarities and differences among evolutionarily related OMBBs. To do this, we need (1) structural information and (2) a sensitive alignment method. With respect to structural information, we need an OMBB’s strand number and strand/loop boundaries. With respect to the alignment method we need a sensitive method which identifies evolutionarily related OMBBs and which accurately matches related parts. For structural information, the accuracy of structure prediction is limited. A recent study reports 76% accuracy in correct topology prediction (Tsirigos et al., 2016). While this accuracy is impressive, it is insufficient for our purposes. To address our structural needs, we used OMBBs with experimentally-solved structures, so that we have the true strand number and strand register.”

And to the Discussion section:

“Although using only structurally-solved proteins may introduce some bias in the dataset, high resolution structures are necessary to find these detailed evolutionary pathways.”

2) The author's claims highly depend on the similarity between proteins/strands, which highly depends on the cut-off E-value of the alignments. This cut-off determines how significant the found patterns are. Since the paper says most β-barrel proteins are homologous (Introduction), I would expect a very low E-value cut-off should have been used in the study. Instead, the paper uses some relatively large values (10^-1^~10^-3^). Certain statistic tests or at least some explanations are required to justify the usage of such E-value cut-off.

As explained above, we remade the entire network using the more widely available uniclust30_2017_10 database for generating the HMMs.

The current manuscript exclusively uses these new alignments. In addition to making our work more reproducible, the uniclust database also generated a more highly connected network. Connections that we previously only saw at an E value of 1 we now found at E value < 10^-3^.

The current manuscript uses no E-values higher than 10^-3^. Reassuringly, the new calculations recapitulate the observations reported previously, but with stronger statistical power.

3) The OMBBs with 22 strands do not seem to be evolutionary related to other OMBBs at lower e-values. Even at e-value of 10^-2^, a large number of 22 OMBBs do not align with 14-stranded OMBBs. How can the 22 strands BB evolve so fast that they no longer can align with 14 strand OMBBs? Supposedly the 14 to 22 strand evolutionary event is a relatively new one.

Reassuringly, the new data presented in the revised manuscript finds the 22-stranded β-barrels to be related to the 14-stranded β-barrels at a much lower E-value of 10^-5^. Nevertheless, we cannot address the specific question, and we admit to it in subsection “Transition from 14- to 22-stranded barrels”.

4) The paper finds some interesting patterns that connect β-barrel proteins with different sizes together. However, all the results shown look like case study, and few statistics are reported to show how prevalent and significant these patterns are in this protein population.Here I list only one example to show this issue. In subsection “Internal repeats”: "The 12-, 14-, 18-, and 22-stranded barrels have single hairpin shifts. The 8- and 16- stranded barrels sometimes have double hairpin shifts and sometimes have single hairpin shifts." What are the percentages of the 12-, 14-, 18-, and 22-stranded barrels that have single hairpin shifts and what is the "sometimes" for 8- and 16-stranded barrels are not clear. Concrete numbers should be reported instead of descriptive language.

We appreciate this concern and we regret the use of descriptive non-statistical language in the text. We have added concrete numbers in all cases. For example:

“…We find a C-terminal alignment in 166/186 alignments involving an 8-stranded barrel.”

“…Essentially all of these (31/32 cases representing relationships among 14,306 sequences in the HMMs) follow the pattern of C-terminal alignment observed in step 1.”

“…Strands 1-3 align with strands 5-7 in 8/14 of 8-stranded barrels.

The alignments from the 16-stranded barrels to 18-stranded barrels show extensive variation (Figure S3I), falling into four broad categories. The largest category (44% of the alignments observed at E-value < 10^-3^, representing relationships among 7,902 sequences in the HMMs) are a C-terminal alignment of two to 10 strands in length, similar to that observed in step 1 with the 8-stranded barrels. The second-largest category (32% of the alignments, representing relationships among 8,618 sequences in the HMMs) are an N-terminal alignment in which the first strands align, the second strands align, etc., with between two and 16 strands involved. In the third-largest group, 14% of the alignments (representing relationships among 5,882 sequences in the HMMs) have strands 1-14 of the 16-stranded barrel aligning with strands 4-18 of the 18-stranded barrel. Finally, in the fourth and smallest group 26 alignments (10%) have strands which align with loops or loops which align with strands. We categorize these alignments as (A) loop to hairpin, (B) loop to hairpin with alternate alignment, and (C) large rearrangement (Figure 5).”

5) What mechanism was responsible for events that do not involve complete gene duplication? e.g. 14 to 22 strands. The authors should consider mapping the TM strands on to the exons and explore if exon duplication might be one of the mechanisms that achieves this?

We have added more discussion of the 14- to 22-stranded transition as follows:

“The mechanism of amplification for the 14- to 22-stranded barrel transition remains opaque. It may be that barrels larger than 18 strands require a stabilizing plug. The 14-stranded barrels from which the 22-stranded barrels evolved have small plug domains; however, the 18-stranded barrels do not have plugs. The presence of the plug could have facilitated the 22-stranded barrels to branch off of the 14-stranded barrels. However, the unusual, non-C-terminal localization of the strands from the 14-stranded barrel in the 22-stranded barrel, as well as the overall lack of internal duplication in 22-stranded barrels, implies that this transition used a different style of amplification than the mechanisms described here. We hope that future analysis will explain this puzzling transition.”

However, we cannot explore exon duplication, as prokaryotes do not have them. We have added the following note on this subject to the Introduction to increase clarity:

“… the lack of exons in prokaryotes eliminates the possibility of complex editing and recombination events as a mechanism for increasing biological diversity.”

6) The authors should comment if an event similar to that of converting an extracellular facing strand to a periplasmic-facing strand has been observed in α helical membrane proteins.

This is a good point. Unlike β-barrels, which we show to have extremely rigid topoplogy, α-helical membrane proteins do occasionally exist with dual-topology. Specifically, they insert into the membrane in both directions and are usually functional as antiparallel oligomers. We have added a brief comment and a reference for readers who want to know more about this subject.

“… The topological rigidity of duplications in β-barrels contrasts with duplications in helical proteins, which are distributed roughly evenly between parallel and antiparallel duplications (Hennerdal et al., 2010).”

7) The writing needs a lot of improvement. The English of the paper is neither plain nor scientific. Just one example (subsection “Step 1”) to show this:This manifests as "at least some" of the 8 strands of the 8-stranded barrels "almost always" (150/151 cases) aligning with "at least some" of last eight strands of the 10-, 12-, and 16-stranded barrels such that strand 1-"when it aligns"-aligns to the 8th to the last strand, and strand 2 to the 7th to last strand, 3 to the 6th to last strand, "etc.".A sentence in a scientific paper should not spans four rows accompanied with a lot of vague terms (shown in quotes above) but lacking concrete statistic numbers, can make any point clear. And this is not an isolated case.

We sincerely apologize to the reviewer for the poor writing. In addition to the removal of all descriptive terms detailed in comment 4 we have employed a professional editor to increase readability. We hope you find this version improved.

[Editors' note: the author responses to the re-review follow.]

Essential revisions:There was discussion among the reviewers regarding the extent to which including predicted secondary structure information would improve the analysis. Reports of secondary structure prediction as high as 87% have been reported (Ou et al., 2010). It’s unclear to what extent the assignments of evolutionary relationships would be sensitive to precise strand boundary definitions. We would like the authors to consider this question and justify the choice of eliminating sequences where structural data are not available.

We appreciate this concern; below we address the three questions raised, one by one.

a) There was discussion among the reviewers regarding the extent to which including predicted secondary structure information would improve the analysis. Reports of secondary structure prediction as high as 87% have been reported (Ou et al., 2010).

We use secondary structure information that was determined experimentally, rather than relying on predicted information, because we think this is key to a meaningful analysis. OMBB topology predictors are only mostly correct – for example, BOCTOPUS2 (Hayat et al., 2016), which is considered the current state of the art, and superseded Ou et al.'s method reports that they correctly predicted only 69% of the proteins in the test set. Thus, even though we are generally enthusiastic users of BOCTOPUS2, we feel that predictions cannot replace experimental data in a detailed analysis. (On a practical level, we cannot use the webserver of Ou et al., (https://rbf.bioinfo.tw/~sachen/BARRELpredict/TMBETAPRED-RBF.php) because it is not maintained.) Consequently, when generating strand-based evolutionary models, using predicted structures where there is no certainty of the strand number or identity would make our results less believable, not more.

The text has been amended to say:

“For structural studies, the accuracy of structure prediction is limited. A recent study reports 69% accuracy in correct topology prediction (Hayat et al. 2016). While this accuracy is impressive, it is insufficient for our purposes. Therefore, to address our structural needs we used OMBBs with experimentally-solved structures, so that we have the true strand number and strand identity.”

b) It’s unclear to what extent the assignments of evolutionary relationships would be sensitive to precise strand boundary definitions.

We appreciate this point and we discuss this consideration extensively below in #3 of “Other points to be addressed”. In brief, we find below that our results are not sensitive to slight shifts in the extracellular region between strand and loop. However, we anticipate that our results would be very sensitive to misidentified strands and incorrect strand numbers.

c) We would like the authors to consider this question and justify the choice of eliminating sequences where structural data are not available.

We appreciate the opportunity to clarify this point. We have not eliminated sequences when structural data are not available; we maintain them as part of the HMMs. In total, our HMMs are derived from 50,832 OMBB homologues. ounts of the number of sequences of likely OMBBs of proteobactera has been assessed at 48,731 (Freeman and Wimley, 2012) and 76,760 (Reddy and Saier, 2016). Given the relative scale of the sequences used in our HMMs compared to the total number of possible sequences which could be used, our dataset has good coverage of the sequences for which structural data are not available.

We have updated our manuscript to make the point more strongly:

“Moreover, the 97 experimentally solved structures of prototypical barrels in our dataset map to 50,832 homologous sequences incorporated into our HMM alignments. Given that the total number of OMBB sequences was assessed to be 48,731 (Freeman and Wimley, 2012) or 76,760 (Reddy and Saier, 2016), our methods have good coverage of the space of the non-structurally characterized OMBBs.”

Other points to be addressed:1) "E-values are always lower among barrels of the same strand number than between barrels of a different strand number." Can the authors be sure that this is due purely to similarity, or is there a concern that either the cost of inserting gaps or the greater number of expected alignments adding to noise when using a shorter query to probe a longer target? This may be accounted for in the analysis, but it was not obvious how it was handled. Related to this, is it possible whether the lack of connections from the largest barrels (18- and 22-stranded) to the smallest (8, 10, 12-stranded), might reflect challenges to the scoring schemes when such length differentials are present. How sensitive are the results for these particular connections to lowering of the thresholds? A comment as to the reasons for choosing E <= 10^-3^ for selection of the cases in the prototypical barrel group (Results section), would be helpful.

The reviewers bring up an important point that we regret we did not make clear enough in the manuscript. Specifically, that when shorter proteins are aligned with longer proteins the shorter protein is not stretched to align with the entire long protein. Rather, the piece of the longer protein that is similar to the shorter protein is aligned to the shorter protein (a so-called local alignment). For the HHSearch algorithm there is no more likelihood for a short piece of protein to have gaps when aligning to a long piece of protein than there is likelihood that there will be gaps between a long piece aligned with a long piece. The fit is not forced to take up the entire protein and all alignments are based on the particular length that is similar. In order to avoid nonsensically short alignments, we filter out any alignment that is less than 20 amino acids (the size of a hairpin).

To clarify this, we have added the following to the manuscript:

“HHSearch uses local alignments, meaning that it is not forced to align full sequences. Rather it can align subsections within proteins and can identify more than one possible alignment. We applied a 20-residue cutoff which is a length that includes 99.7% of hairpins in our database.”

In order to describe the reason for using the 10^-3^ E-value cutoff we have added the following to the manuscript:

“In this paper, we focus exclusively on the largest group found at an E-value of 10^-3^, which we call the prototypical β-barrels. We use the E-value ≤ 10^-3^ because that value is the most widely used to infer homology, i.e., evolution from a common ancestor, (Pearson, 2013) and because it cleanly distinguished the prototypical barrels from other functional groups. The other groups of OMBBs, such as the efflux pumps, assembly barrels, and Fim/usher proteins are described more fully elsewhere (Franklin et al., 2018a).”

2) The conservation of 8 strands throughout the barrel families (depicted in Figure 4) is fascinating, particularly because one would expect that larger barrels would require larger amino acids to enforce the curvature of the barrel circumference. Since the authors know the register of the HMM profiles with respect to structure, can it be said whether this conservation is due to core vs surface residues? Either way – it would inform the discussion regarding the mechanism of folding. A conserved folding nucleus would imply core facing residues are conserved. A conserved surface would imply a different transition state in the presence of the BamA machinery. Of course, both could be true.

Since core residues and surface residues alternate one after the other and our alignments are not for individual amino acids, but rather for protein fragments of at least 20 amino acids, we cannot assess if core or surface residues are more conserved from our data. However, there is already a very thorough analysis on this topic showing that the outward-facing rate of substitution is twice as large as the rate of substitution for inward-facing amino acids (Jimenez-Morales and Liang, 2011).

At the reviewers’ suggestion we have assessed if side chain volume increases with barrel strand number. The reviewers are correct in that there is an increase in amino acid volume for both inward and outward facing residues. We have added this to the supplement and added the following note in the manuscript:

Overall, we find that as barrels increase in strand number there is a slight increase in the average amino acid volume in the strands (Figure 2—figure supplement 1), likely because larger barrels would require larger amino acids to enforce the smaller curvature of the larger barrel’s circumference. However, such differences do not mute the relationships between barrels of different sizes.

3) The authors are using 3D structure of the proteins to gather the strand start and strand register values. It is not clear to how the authors address the artifacts introduced in the X-ray structures. For example, crystals for BamA from Salmonella and Haemophilus ducreyi have been resolved (5ORI and 4K3C respectively). Strands 1 and 16 in both structures have significantly different lengths (3 and 5 in 5ORI and 8 and 10 in 4K3C). How is this resolved?

We appreciate this concern. Although the particular example of BamA is not applicable to our dataset because the assembly barrels are not part of the prototypical barrel group, there are many barrels within the prototypical barrels that have alternate structures with different strand boundaries than the boundaries of the crystal structures reported. OMBBs fray at the extracellular end. Meaning they do not always have strict strand boundaries in the dynamic extracellular region. It is frequently the case that a few amino acids defined as the top of a strand in one structure will be defined as the bottom of a loop in another structure.

In general, we find crystal structures typically have much longer strands than NMR structures because of the dynamics of the extracellular region. We also find that there are alternate crystal structures which do not have sufficient electron density in the dynamic loop/strand region and so the structure is not resolved at all in that region. We have reviewed all of our analyses with the longest and shortest definitions of strands and find only one alignment, which includes a different number of strands when alternate structures are used. This alternate alignment does not change any of the results reported.

We have added the following to the text:

“Some barrels have alternative structures in which some amino acids in the extracellular region of the barrel are defined as loop where in other structures they are defined as strand or vice versa. Barrels for which multiple structures are available, were analyzed using the longest and shortest strand definitions. Using alternative structures alters the number of strands in only one of the 1653 alignments in the 10^-3^ E-value network (Figure 5—figure supplement 2). That alignment does not change the features reported here nor is it one of the transitions described in Figure 5.”

4) In "Transition from 14- to 22- stranded barrels", the authors claim the 18- stranded barrels do not have plugs. This is incorrect, most barrels > 10 strands have in-plugs.

The reviewers are correct, most barrels with > 10 strands have some proteinacious content inside the pore, usually a helix or a disordered helix. The 14- and 22- stranded barrels have domains with tertiary structure plugging their pores and it was this that we were trying to bring attention to.

We have clarified this in the text as follows:

“It may be that barrels larger than 18 strands require a large stabilizing domain plugging the pore, such as the β-sandwich observed in the middle of the 22-stranded barrels. The 14-stranded barrels from which the 22-stranded barrels evolved have small N-terminal plug domains with tertiary structure; other barrels with more than 10 strands have occluding loops but not domains with tertiary structure. The presence of the more structured plug could have facilitated the 22-stranded barrels to branch off of the 14-stranded barrels.”

5) Why are the barrels > 22 strands not included in the study (PapC, 2vqi; FimD, 3rfz; LptD, 4q35)?

Like the assembly barrels discussed in “points to be addressed” #2 the Fim/Usher proteins are not strongly related to the other prototypical barrels and are not included in this study. The groups that are not in the prototypical barrels are more completely described in our previously published paper (Franklin et al., 2018a)

In order to address this question in the manuscript, we have added the following note in the manuscript:

“The other groups of OMBBs, such as the efflux pumps, assembly barrels, and Fim/usher proteins are described more fully elsewhere (Franklin et al., 2018a).”

6) For the average polarity and hydrophobicity of the loops and strands, it would be useful to have a control in a different region of the protein, as well as standard deviations to get a sense of the variability of these values in context.

This is a great suggestion. We have created separate loop and strand datasets from the loops and strands of the 16- and 18- stranded barrels that were not involved in loop-strand transitions. We write:

“Control datasets for both loops and strands were created using all the loops and strands in 16- and 18- stranded β-barrels that were not part of any loop-strand transition alignment.”

Comparing our loops and strands of interest to the controls we find the following as described in the manuscript:

“Β-strands have a sequence hallmark of alternating polar and non-polar amino acids. Therefore, we checked for an increase in polarity alternation between the residues which are part of the alignments that are loops in one structure and strands in another structure. The average polarity alternation in the loop conformation is 45% and the average polarity alternation in the strand conformation is 55%. However, both polarity alternation percentages are more like the alternation percentage of our control set of strands (44%) than the alternation percentage of our control loops (28%).”

In contrast to the similarity in polarity alternation between loops and strands, we found that the loop regions were less hydrophobic than the strand regions with which they aligned although they are more hydrophobic than the loops in our control. We averaged hydropathy values (Kyte and Doolittle, 1982) for the amino acids which are part of the alignments and are loops in one structure and strands in another structure. The average hydropathy in the loop conformation is -0.57 and the average hydropathy in the strand conformation is -0.22 (negative is hydrophilic and positive is hydrophobic). Though these differ from each other, they are more hydrophobic and less different than our control loops and strands which we found to have hydropathy scores of -1.11 and -0.40 respectively.

7) It would be best if the in-house software Polar Bearal (subsection “Polarity alternation and hydrophobicity calculations”) were made available through GitHub or a similar portal.

We agree with the reviewers and have made Polar Bearal available on GitHub.

The manuscript has been updated to say:

Strands were defined using in-house software, Polar Bearal, as previously described (Slusky and Dunbrack, 2013; Franklin et al., 2018b), available from https://github.com/SluskyLab/PolarBearal.git.